# CASA: Bridging the Gap between Policy Improvement and Policy Evaluation with Conflict Averse Policy Iteration

## Abstract

We study the problem of model-free reinforcement learning, which is often solved following the principle of Generalized Policy Iteration (GPI). While GPI is typically an interplay between policy evaluation and policy improvement, most conventional model-free methods with function approximation assume the independence of GPI steps, despite of the inherent connections between them. In this paper, we present a method that attempts to eliminate the inconsistency between policy evaluation step and policy improvement step, leading to a conflict averse GPI solution with gradient-based functional approximation. Our method is capital to balancing exploitation and exploration between policy-based and value-based methods and is applicable to existing policy-based and value-based methods. We conduct extensive experiments to study theoretical properties of our method and demonstrate the effectiveness of our method on Atari 200M benchmark.

## 1 Introduction

Model-free reinforcement learning has made many impressive breakthroughs in a wide range of Markov Decision Processes (MDP) (Vinyals et al., 2019; Pedersen, 2019; Badia et al., 2020). Overall, the methods could be cast into two categories, value-based methods such as DQN (Mnih et al., 2015) and Rainbow (Hessel et al., 2017), and policy-based methods such as TRPO (Schulman et al., 2015), PPO (Schulman et al., 2017) and IMPALA (Espeholt et al., 2018).

Value-based methods learn state-action values and select the action according to their values. The main target of value-based methods is to approximate the fixed point of the Bellman equation through the generalized policy iteration (GPI) (Sutton & Barto, 2018), which generally consists of policy evaluation and policy improvement. One characteristic of the value-based methods is that unless a more accurate state-action value is estimated by iterations of the policy evaluation, the policy will not be improved. Previous works equip value-based methods with many carefully designed structures to achieve more promising reward learning and sample efficiency (Wang et al., 2016; Schaul et al., 2015; Kapturowski et al., 2018).

Policy-based methods learn a parameterized policy directly without consulting state-action values. One characteristic of policy-based methods is that they incorporate a policy improvement phase in every training step, while in contrast, the value-based methods only change the policy after the action corresponding to the highest state-action values is changed. In principle, policy-based methods perform policy improvement more frequently than value-based methods.

We notice that value-based and policy-based methods locate at the two extremes of GPI, where value-based methods won't improve the policy until a more accurate policy evaluation is achieved, while policy-based methods improve the policy for every training step even when the policy evaluation hasn't converged. To

mitigate the defect of each, we pursuit a technique that is capable of balancing between the two extremes flexibly. We first study the gradients between policy improvement and policy evaluation and notice that they show a positive correlation statistically during the entire training process. To find out if there exists a way that the gradients of the policy improvement and the policy evaluation are parallel, we propose CASA, **Critic AS** an **A**ctor, which satisfies a weaker compatible condition (Sutton et al., 1999) and enhances gradient consistency between policy improvement and policy evaluation.

With further delving into the properties of CASA, we find CASA is an innovative combination of value-based and policy-based methods. When the policy-based methods are equipped with CASA, the collapse to the sub-optimal solution as the entropy goes to zero is prevented by the evaluation of the state-action values, which encourages exploration. When the value-based methods are equipped with CASA, the policy improvement via policy gradient is equivalent to the evaluation of the state-action values and a self-bootstrapped policy improvement, which enhances exploitation.

To enable CASA for a large scale off-policy learning, we introduce Doubly-Robust Trace (DR-Trace), which exploits doubly-robust estimator (Jiang & Li, 2016) and guarantees the synchronous convergence of the state-action values and the state values.

Our main contributions are as follows:

(i) We present a novel method CASA which enhances gradient consistency between policy evaluation and policy improvement and present extensive studies on the behavior of the gradients.

(ii) We demonstrate CASA could be freely applied to both policy-based and value-based algorithms with motivating examples.

(iii) We present extensive empirical study on Atari benchmark , where our conflict averse algorithm brings substantial improvements over the baseline methods.

## 2 PRELIMINARY

Consider an infinite-horizon MDP, defined by a tuple $(\mathcal{S}, \mathcal{A}, p, r, \gamma)$, where $\mathcal{S}$ is the state space, $\mathcal{A}$ is the action space, $p : \mathcal{S} \times \mathcal{A} \times \mathcal{S} \to [0, 1]$ is the state transition probability function, $r : \mathcal{S} \times \mathcal{A} \to \mathbb{R}$ is the reward function, and $\gamma$ is the discounted factor. The policy is a mapping $\pi : \mathcal{S} \times \mathcal{A} \to [0, 1]$ which assigns a distribution over the action space given a state.

The objective of reinforcement learning is to maximize the *return*, or cumulative discounted rewards,

$$\text{maximize } \mathcal{J} = \mathbb{E}_{traj \sim \pi}\left[\sum_t \gamma^t r(s_t, a_t)\right], \tag{1}$$

where $traj = \{s_0, a_0, r_0, \dots\}$ is a trajectory sampled by $\pi$ with policy-environment interaction.

Value-based methods maximize $\mathcal{J}$ by estimating various type of value functions: the state value function is defined as $V^\pi(s) = \mathbb{E}_\pi\left[\sum_t \gamma^t r_t | s_0 = s\right]$, the state-action value function is defined as $Q^\pi(s, a) = \mathbb{E}_\pi\left[\sum_t \gamma^t r_t | s_0 = s, a_0 = a\right]$; the advantage function is defined as $A^\pi(s, a) = Q^\pi(s, a) - V^\pi(s)$. The objective of maximizing the value functions in value-based methods can be improved through GPI until converging to the optimal policy. For the approximated state-value function $Q_\theta$ that estimates $Q^\pi$, the policy evaluation is conducted by:

$$\text{minimize } \mathbb{E}_\pi[(Q^\pi(s, a) - Q_\theta(s, a))^2], \tag{2}$$

where $Q^\pi$ is estimated by various methods, e.g., $\lambda$-return (Sutton, 1988) and ReTrace (Munos et al., 2016). The policy improvement is usually achieved by greedily selecting actions with the highest state-action values.

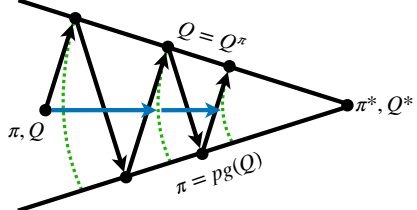

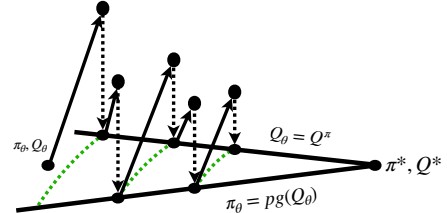

Figure 1: The GPI process in our work. Unlike (Sutton & Barto, 2018), we evaluate $\pi$ by $Q$ instead of $V$, and we improve $\pi$ using policy gradient ascent ($pg$ for brevity) instead of greedy. The learning procedure is shown by the black arrows, i.e., $\mathbf{E} \to \mathbf{I} \to \mathbf{E} \to \mathbf{I} \cdots$.

Figure 2: GPI with function approximation. Due to the constraint of approximated function space, the ideal policy iteration cannot be actually achieved. The underlying process of GPI with function approximation can be regarded as doing policy improvement and policy evaluation in an ideal space then being projected back into the approximated function space (Sutton & Barto, 2018; Ghosh et al., 2020).

Policy-based methods maximize $\mathcal{J}$ by optimizing some parameterized policy $\pi_\theta$ according to the policy gradient theorem (Sutton & Barto, 2018),

$$\nabla_\theta \mathcal{J} = \mathbb{E}_\pi[\Psi(s, a)\nabla_\theta \log \pi_\theta(a|s)]. \tag{3}$$

The vanilla policy gradient uses $\Psi = \sum_{t=0}^\infty \gamma^t r_t$. Actor-critic algorithms approximate $\Psi(s, a)$ in the form of baseline, e.g., IMPALA (Espeholt et al., 2018) adopts $\Psi(s, a) = r + \gamma V^{\tilde\pi}(s') - V_\theta(s)$ and uses V-Trace to estimate $V^{\tilde\pi}$.

## 3 METHODOLOGY

### 3.1 MOTIVATION

We use $V_\theta$ to estimate $V^\pi$, $Q_\theta$ to estimate $Q^\pi$ and $\pi_\theta$ to represent the policy, where $\theta$ represents all parameters to be optimized. In this work, there is one backbone and two individual heads after the backbone. The advantage function and the policy share one head, and state value function is the other head. Hence the policy reuses all parameters of value functions except that temperature $\tau$ is only for the policy. We keep $\tau$ static in this work. We use $\mathbf{E}$ to represent the policy evaluation, which gives the ascent direction of the gradient by $\theta \leftarrow \theta + \eta \mathbb{E}_\pi[(Q^\pi - Q_\theta)\nabla_\theta Q_\theta]$. We use $\mathbf{I}$ to represent the policy improvement, which gives $\theta \leftarrow \theta + \eta \mathbb{E}_\pi[(Q^\pi - V_\theta)\nabla_\theta \log \pi_\theta]$.

Let's recap the GPI process as shown in Figure 1. To get rid of the function approximation error, we first assume the approximation function enjoys infinite capacity. We use $< x, y >$ to denote the angle between two vectors, where $< x, y >= \arccos(\frac{x \cdot y}{||x|| \cdot ||y||})$ with $\arccos : [-1, 1] \to [0, \pi]$. We define an important notion $\beta$, which represents the angle between the gradient ascent directions of $\mathbf{I}$ and $\mathbf{E}$, as follows,

$$\beta \overset{def}{=} < \mathbb{E}_\pi[(Q^\pi - Q_\theta)\nabla_\theta Q_\theta], \mathbb{E}_\pi[(Q^\pi - V_\theta)\nabla_\theta \log \pi_\theta] > . \tag{4}$$

When $\beta = 0$ i.e. $\cos(\beta) = 1$, $\mathbf{I}$ and $\mathbf{E}$ become parallel to each other, which is the blue arrow in Figure 1, and there is no conflict between the gradient ascent directions of $\mathbf{I}$ and $\mathbf{E}$ anymore. When $\beta = \pi/2$ i.e. $\cos(\beta) = 0$, $\mathbf{I}$ and $\mathbf{E}$ are perpendicular. When $\beta = \pi$ i.e. $\cos(\beta) = -1$, $\mathbf{I}$ and $\mathbf{E}$ are toward exactly opposite directions.

Next, we assume the representation capacity of the approximation function is limited. When the function approximation is involved, i.e. $Q^\pi$ is estimated by $Q_\theta$ and $\pi$ is approximated by $\pi_\theta$, from the view of operators (Ghosh et al., 2020), each of $\mathbf{I}$ and $\mathbf{E}$ can be further decomposed into two operators, as shown in Figure 2. One is to do the policy improvement and the policy evaluation, the other is to project into the restricted function space. When $\beta > 0$, GPI with function approximation would involve two projection

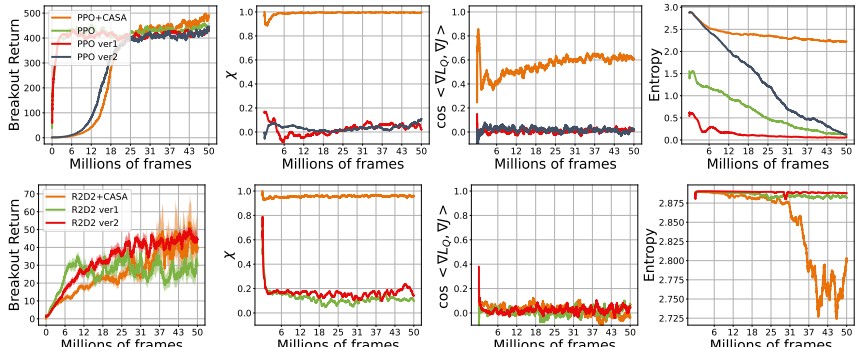

Figure 3: *Return*, $\chi$, $\cos(\beta)$ and *entropy*. PPO is adjusted with two additional versions to evaluate state-action values. R2D2 uses a surrogate policy to approximate policy gradient. Entropy of R2D2 is entropy of Boltzmann policy on state-action values. Details are in Appendix B.

operators in each iteration, which introduces inevitable approximation error. When $\beta = 0$, if the function approximation error is not considered, we find that the gradient conflict between **I** and **E** would be totally eliminated. If we consider the limitation of the approximation function, similar to the blue arrow in Figure 1, one iteration (represented by two black arrows and two dotted arrows) can be united into one arrow and one dotted arrow (not shown in Figure 2 but analogy to the blue arrow in Figure 1), where the gradient conflict is eliminated and the two projection operators are reduced to one correspondingly.

As stated above, if $\beta = 0$ holds, we can expect that the gradient conflict between the policy improvement and the policy evaluation is eliminated and the function approximation error could be reduced. However, $\beta$ is usually estimated by sampling with stochasticity. It's difficult to let $\beta = 0$ by optimizing $\theta$. Instead, we consider another notion $\chi$ by removing step sizes and taking expectation outside, where the angle of each state is fully controllable by $\theta$.

$$\chi \stackrel{def}{=} \mathbb{E}_\pi[\cos < \nabla_\theta Q_\theta, \nabla_\theta \log \pi_\theta >]. \tag{5}$$

In fact, $\chi$ is highly correlated to compatible value function (Sutton et al., 1999), and Theorem 3 shows that $\chi = 1$ is the necessary condition for the compatible condition $\nabla_\theta Q_\theta = \nabla_\theta \log \pi_\theta$, which is a weaker compatible condition. More details about compatible value function are in Appendix A.

To further understand the behavior of $\beta$ and $\chi$, we track $\cos(\beta)$ and $\chi$ of two algorithms PPO and R2D2 as representatives for policy-based and value-based methods, respectively. We show an important fact in Figure 3 that both $\chi$ and $\cos(\beta)$ are statistically positive for both original version and adjusted versions, which means that $\arccos(\chi)$ and $\beta$ are likely to be less then $\pi/2$ with neural network approximated functions. The aforementioned conceptual and empirical findings inspire us to raise the following question on GPI: whether we can guarantee $\chi = 1$, so that $\cos(\beta)$ is also closer to 1.

## 3.2 FORMULATION

Denote $\tau \in \mathbb{R}_+$ to be a positive temperature and $sg$ to be a *stop gradient* operator. CASA can estimate $V_\theta$ and $A_\theta$ by any function parameterized by $\theta$, where $\pi_\theta$ and $Q_\theta$ are derived as follows:

$$\begin{cases} \pi_\theta(\cdot|s) = \text{softmax}(A_\theta(s, \cdot)/\tau), \\ \bar{A}_\theta(s, a) = A_\theta(s, a) - \sum_{a'} sg(\pi_\theta(a'|s))A_\theta(s, a'), \\ Q_\theta(s, a) = \bar{A}_\theta(s, a) + sg(V_\theta(s)). \end{cases} \tag{6}$$

Note that there exist two $sg$ operators in equation 6. The first $sg$ operator is used for computing advantage as $\bar{A}_\theta = A_\theta - \mathbb{E}_\pi[A_\theta] = A_\theta - sg(\pi_\theta) \cdot A_\theta$, where the $sg$ operator here guarantees the gradients between

policy improvement and policy evaluation are parallel, which we elaborate later. Intuitively, this $sg$ operator also means that we keep $\pi_\theta$ unchanged when evaluating the policy $\pi_\theta$. The second $sg$ operator exists in $Q_\theta = \bar{A}_\theta + sg(V_\theta)$. As (Chen & He, 2020) regards $sg$ in siamese representation learning as a case of EM-algorithm (Dempster et al., 1977), a similar interpretation exists here. $Q_\theta = \bar{A}_\theta + sg(V_\theta)$ decomposes the estimation of $Q_\theta$ into a two stage problem, where the first is to estimate the advantage of each action without changing the expectation, the second is to estimate the expectation.

The equation 6 includes a straightforward refinement of dueling-DQN. We know dueling-DQN estimates $Q^\pi$ by $Q_\theta = A_\theta + V_\theta$, but it cannot guarantee $\mathbb{E}_\pi[A_\theta] = 0$ i.e. $\mathbb{E}_\pi[Q_\theta] = V_\theta$ due to the function approximation error. But if we estimate $Q_\pi$ by $Q_\theta = A_\theta - \mathbb{E}_\pi[A_\theta] + V_\theta$, it satisfies the necessary condition $\mathbb{E}_\pi[Q_\theta] = V_\theta$ without loss of generality.

### 3.3 PATH CONSISTENCY BETWEEN POLICY EVALUATION AND POLICY IMPROVEMENT

For brevity, we omit $\theta$ and $V, Q, A, \pi$ are all approximated functions. Denote the estimations of $V$ and $Q$ as $V^{\tilde{\pi}}$ and $Q^{\tilde{\pi}}$ respectively. For instance, one choice is to calculate $V^{\tilde{\pi}}$ and $Q^{\tilde{\pi}}$ by V-Trace (Espeholt et al., 2018) and ReTrace (Munos et al., 2016) respectively.

At training time, the policy evaluation is achieved by updating $\theta$ to minimize,

$$L_V(\theta) = \mathbb{E}_\pi[(V^{\tilde{\pi}} - V)^2], \ L_Q(\theta) = \mathbb{E}_\pi[(Q^{\tilde{\pi}} - Q)^2],$$

which gives the ascent direction of $\theta$ by:

$$\nabla_\theta L_V(\theta) = \mathbb{E}_\pi\left[(V^{\tilde{\pi}} - V)\nabla_\theta V\right], \ \nabla_\theta L_Q(\theta) = \mathbb{E}_\pi\left[(Q^{\tilde{\pi}} - Q)\nabla_\theta Q\right]. \tag{7}$$

And we make the policy improvement by policy gradient, which gives the ascent direction of $\theta$ by:

$$\nabla_\theta \mathcal{J}(\tau, \theta) = \mathbb{E}_\pi\left[\tau(Q^{\tilde{\pi}} - V)\nabla_\theta \log \pi\right], \tag{8}$$

where $\mathcal{J}(\tau, \theta) = \tau \mathbb{E}_\pi[\sum \gamma^t r_t]$. It takes an additional $\tau$, which frees the scale of gradient from $\tau$.

The final gradient ascent direction of $\theta$ is given by:

$$\alpha_1 \nabla_\theta L_V + \alpha_2 \nabla_\theta L_Q + \alpha_3 \nabla_\theta \mathcal{J}. \tag{9}$$

With $(V, Q, \pi)$ defined in equation 6, by Lemma E.1, we have,

$$\nabla_\theta Q = (\mathbf{1} - \pi)\nabla_\theta A = \tau \nabla_\theta \log \pi. \tag{10}$$

For brevity, denote the shared gradient path as $\mathbf{g} = (\mathbf{1} - \pi)\nabla_\theta A$.

Plugging equation 10 into equation 7 equation 8, we have,

$$\nabla_\theta L_Q = \mathbb{E}_\pi\left[(Q^{\tilde{\pi}} - Q)\mathbf{g}\right], \nabla_\theta \mathcal{J} = \mathbb{E}_\pi\left[(Q^{\tilde{\pi}} - V)\mathbf{g}\right]. \tag{11}$$

By equation 11, $\nabla_\theta L_Q$ and $\nabla_\theta \mathcal{J}$ walk along the same vector direction of gradient path $\mathbf{g}$ for each state. By equation 10, this is exactly the case $\chi = 1$. Since all parameters to estimate $Q$ and $\pi$ are shared except for $\tau$, we call it **C**ritic **AS** an **A**ctor.

If we make a subtraction between $\nabla_\theta L_Q$ and $\nabla_\theta \mathcal{J}$, we have,

$$\nabla_\theta \mathcal{J} = \nabla_\theta L_Q + \mathbb{E}_\pi\left[(Q - V)\mathbf{g}\right]. \tag{12}$$

We know $\mathbb{E}_\pi\left[(Q - V)\mathbf{g}\right]$ is a self-bootstrapped policy gradient with function approximated $Q$. Recalling the fact that the value-based methods improves the policy by greedily selecting actions according to $Q$, if we apply $\nabla_\theta \mathcal{J}$ on $\theta$, it additionally utilizes $Q$ to do policy improvement. This is a more greedy usage of $Q$ to improve policy than its usual usage.

If we exploit the structural information as $(V, Q, \pi)$ defined by equation 6, by Lemma E.2,

$$\mathbb{E}_\pi\left[(Q-V)\mathbf{g}\right] = \tau\mathbb{E}_\pi\left[(Q-V)\nabla_\theta \log \pi\right] = -\tau^2\nabla_\theta\mathbf{H}[\pi],$$

then we have,

$$\nabla_\theta L_Q = \nabla_\theta\mathcal{J} + \tau^2\nabla_\theta\mathbf{H}[\pi]. \tag{13}$$

The equation 13 shows $\nabla_\theta L_Q$ is a policy gradient with an entropy regularization. If we apply $\nabla_\theta L_Q$ on $\theta$ for policy-based methods, an entropy regularization works implicitly by $\alpha_2\nabla_\theta L_Q$ in equation 9, which prevents the policy collapse to a sub-optimal solution.

### 3.4 DR-Trace and Off-Policy Training

| | DR-Trace | V-Trace / ReTrace |
|---|---|---|
| | $\delta_t^{DR}=r_t+\gamma V(s_{t+1})-Q(s_t,a_t)$ | $\delta_t^{V/Q}=r_t+\gamma V(s_{t+1})/Q(s_{t+1},a_{t+1})-V(s_t)/Q(s_t,a_t)$ |
| $V^{\tilde\pi}$ | $\mathbb{E}_\mu[V_t+\sum_{k\geq0}\gamma^k c_{[t:t+k-1]}\rho_{t+k}\delta_{t+k}^{DR}]$ | $\mathbb{E}_\mu[V_t+\sum_{k\geq0}\gamma^k c_{[t:t+k-1]}\rho_{t+k}\delta_{t+k}^{V}]$ |
| $Q^{\tilde\pi}$ | $\mathbb{E}_\mu[Q_t+\sum_{k\geq0}\gamma^k c_{[t+1:t+k-1]}(1_{\{k=0\}}+1_{\{k>0\}}\rho_{t+k})\delta_{t+k}^{DR}]$ | $\mathbb{E}_\mu[Q_t+\sum_{k\geq0}\gamma^k c_{[t+1:t+k]}\delta_{t+k}^{Q}]$ |
| $\nabla\mathcal{J}$ | $\mathbb{E}_\mu[\rho_t(Q_t^{\tilde\pi}-V_t)\nabla\log\pi]$ | $\mathbb{E}_\mu[\rho_t(r_t+V_{t+1}^{\tilde\pi}-V_t)\nabla\log\pi]$ |

Table 1: Comparison between DR-Trace and V-Trace/ReTrace.

To enable off-policy training with behavior policy $\mu$, one choice is to estimate $V^{\tilde\pi}$ and $Q^{\tilde\pi}$ in equation 7 and equation 8 by V-Trace and ReTrace. As CASA estimates $(V, Q, \pi)$, applying Doubly Robust (Jiang & Li, 2016) is feasible and suitable. We propose DR-Trace and find the convergence rate and the fixed point of DR-Trace are the same as V-Trace's according to its convergence proof. For completeness, we provide DR-Trace and its comparison with V-Trace/ReTrace in Table 1. More details are in Appendix D.

## 4 Experiments

### 4.1 Basic Setup

We employ a Learner-Actor pipeline (Espeholt et al., 2018) for large-scale training. Motivation and ablation experiments on PPO and R2D2 don't use LSTM, only experiments on CASA+DR-Trace use LSTM (Schmidhuber, 1997), which is for comparison with other algorithms. We use *burn-in* (Kapturowski et al., 2018) when LSTM is used. All estimated values share the same backbone, which is followed by two fully connected layers for each individual head. We use no intrinsic reward and no entropy regularization in any experiment. We find that using life information can greatly increase the performance of some games. However, to be general, we will not end the episode if life is lost. All hyperparameters are in Appendix F.

For brevity, we denote $\nabla L_V = \mathbb{E}_\pi[(V^\pi-V_\theta)\nabla V_\theta]$, $\nabla L_Q = \mathbb{E}_\pi[(Q^\pi-Q_\theta)\nabla Q_\theta]$ and $\nabla\mathcal{J} = \mathbb{E}_\pi[(Q^\pi-V_\theta)\nabla\log\pi_\theta]$, where expectation is batch-wise average in our implementation. When we write $<a,b>$ with $a,b \in \{\nabla L_V, \nabla L_Q, \nabla\mathcal{J}\}$, we firstly calculate batch-wise averaged gradient of $a$ and $b$, then we calculate the angle in-between. When we write $\cos<\nabla Q, \nabla\log\pi>$ or $\chi$, we mean $\mathbb{E}_\pi[\cos<\nabla_\theta Q_\theta, \nabla_\theta\log\pi_\theta>]$, which firstly calculates element-wise cosines and then takes batch-wise average. To avoid numerical problem, we calculate $\frac{x\cdot y}{||x||\cdot||y||}$ by $\frac{x\cdot y}{\max(||x||,10^{-8})\cdot\max(||y||,10^{-8})}$.

### 4.2 Application of CASA on Representative Algorithms

CASA is applicable to existing algorithms. We take PPO and R2D2 for demonstration. The application of CASA on PPO is straightforward. Applying CASA on R2D2 is impossible as either $\epsilon$-greedy policy

| | PPO | | PPO+CASA | R2D2 | | R2D2+CASA |
|---|---|---|---|---|---|---|
| Func. Approx. | $(V, logit) = (V_\theta, logit_\theta)$ $\pi = \text{softmax}(logit)$ | $\Rightarrow$ | $(V, A) = (V_\theta, A_\theta)$ $\pi = \text{softmax}(A/\tau)$ $\bar{A} = A - sg(\pi) \cdot A$ $Q = \bar{A} + sg(V)$ | $(V, A) = (V_\theta, A_\theta)$ $Q = A + V$ | $\Rightarrow$ | $(V, A) = (V_\theta, A_\theta)$ $\pi = \text{softmax}(A/\tau)$ $\bar{A} = A - sg(\pi) \cdot A$ $Q = \bar{A} + sg(V)$ |
| Gradient | $0.5\nabla L_V + \nabla \mathcal{J}$ | $\Rightarrow$ | $0.5\nabla L_V + \nabla L_Q + \nabla \mathcal{J}$ | $\nabla L_Q$ | $\Rightarrow$ | $0.5\nabla L_V + \nabla L_Q + \nabla \mathcal{J}$ |

Table 2: Examples of applying CASA on policy-based methods (PPO) and value-based methods (R2D2).

or $\arg\max Q$ policy breaks the gradient. This problem is the same as calculating the gradients of policy improvement for value-based methods. We use a surrogate policy $\pi_{surrogate} = \text{softmax}(A/\tau)$, which is discussed in Appendix B. Table 2 summarizes adjustments of function approximations and training gradients.

Since PPO+CASA and R2D2+CASA have the same function approximation, recalling the fact that value-based methods improve the policy when a more accurate evaluation is achieved and policy-based methods improve the policy for every step, we can balance the two flexibly with $\chi = 1$ by $\alpha_1, \alpha_2, \alpha_3$ in equation 9.

In Figure 3, algorithms with CASA show much higher $\cos(\beta)$ and $\chi$. PPO+CASA does more exploration than the original PPO, as the entropy of $\pi$ doesn't easily drop to zero. R2D2+CASA tends to distinct the state-action values, where we use the entropy of $Q$ to measure how greedy the current state-action values are.

### 4.3 BEHAVIOR OF GRADIENTS ON DIFFERENT STRUCTURES

| PPO+CASA | $Q = A_\theta - sg(\pi_\theta) \cdot A_\theta + sg(V_\theta)$ |
|---|---|
| type 1 | $Q = A_\theta - \pi_\theta \cdot A_\theta + sg(V_\theta)$ |
| type 2 | $Q = A_\theta - sg(\pi_\theta) \cdot A_\theta + V_\theta$ |
| type 3 | $Q = A_\theta + sg(V_\theta)$ |
| type 4 | $Q = A_\theta + V_\theta$ |
| type 5 | $Q = Q_\theta$ |

Table 3: Behavior of gradient on different types. Type 1&2 are CASA-like structures, where type 1 removes $sg$ of $\pi$ and type 2 removes $sg$ of $V_\theta$. Type 3&4 are dueling-like structures, where type 3 adds $sg$ to $V$ for dueling-Q and type 4 is dueling-Q. Type 5 uses a new head to estimate $Q_\theta$ separately, which can be considered as an auxiliary task to estimate $Q^\pi$.

Though we show that CASA satisfies $\nabla Q \propto \nabla \log \pi$, which means $\chi = 1$, it's unknown if the structure of CASA is unique. As $Q = A - \mathbb{E}_\pi[A] + sg(V)$ is a direct refinement of dueling-DQN, we try several different structures of PPO+CASA. All settings of estimating state-action values are shown in Table 3. We always use $0.5 \cdot \nabla L_V + \nabla L_Q + \nabla \mathcal{J}$ as the training gradient. We present Breakout and Qbert in Figure 4.

For the sake of clarity, we group PPO+CASA and type 3 as $sg$-$V$ group, type 2 and type 4 as $no$-$sg$-$V$ group. The $sg$-$V$ group has higher $\chi$ and higher $\cos(\beta)$, which is closer to the compatible condition and the consistency between two GPI steps, and $no$-$sg$-$V$ group is always worst than its contrast in $sg$-$V$ group.

PPO+CASA has $\chi = 1$ and the highest $\cos(\beta)$. Type 1 has less returns than PPO+CASA. Hence, when applying a CASA-like structure, stopping the gradient of $\pi$ is always preferred.

Type 5 uses an individual head to estimate $Q^\pi$, which performs the worst. Hence, a well-designed CASA-like or dueling-like structure is always preferred.

By scatter plot and box plot in Figure 4, $\chi$ and $\cos(\beta)$ are positive correlated depending on different structures. This phenomenon answers part of the last question of Section 3.1: for these specific designed structures, $\chi$ and $\cos(\beta)$ show positive correlation.

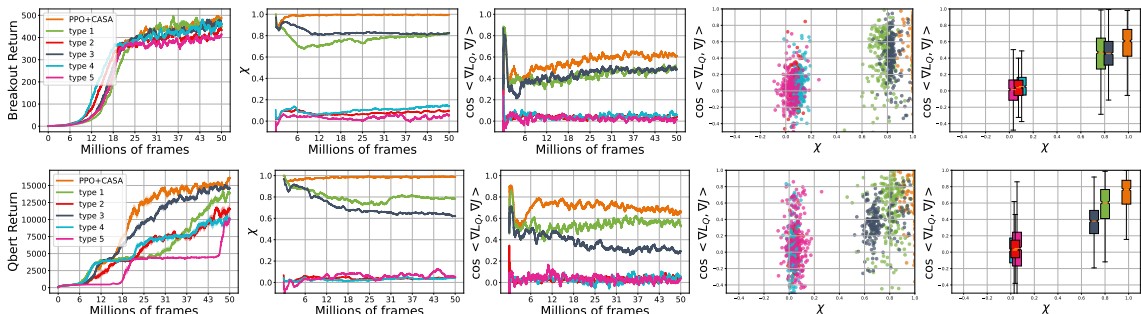

Figure 4: The ablation results evaluated on Breakout (top row) and Qbert (bottom row). From left to right is the *return* curve, $\chi$, $\cos(\beta)$, *scatter plot* of $(\chi, \cos(\beta))$ and *box plot* of $(\chi, \cos(\beta))$. Each scatter point is one batch sampled from every consecutive 100 batches. Each box is the interquartile range of scatter points.

## 4.4 EVALUATION OF CASA ON ATARI GAMES

We present an extensive evaluation on CASA, where we train CASA + DR-Trace on 57 Atari games and report the results in terms of two metrics. The first is Human Normalized Score (HNS), which normalizes the reward by random policy and human expert policies. The other is Standardized Atari BEnchmark for RL (SABER), which normalizes the reward by random policy and human world records, where the normalized score is capped by 200%. SABER is considered because recent studies show that the median HNS could easily get hacked by the algorithm since it is sensitive to improvement on a small subset of games. Table 4 summarizes the results.

|  | Mean HNS | Median HNS | Mean SABER | Median SABER |
|---|---|---|---|---|
| Rainbow | 873.97 | 230.99 | 28.39 | 4.92 |
| IMPALA | 957.34 | 191.82 | 29.45 | 4.31 |
| LASER | 1741.36 | **454.91** | **36.77** | 8.08 |
| **CASA** | **1941.08** | 246.36 | 36.10 | **10.29** |

Table 4: Evaluation scores for the methods on Atari benchmark presented in %.

Note that CASA is a variant of IMPALA with DR-Trace, and it achieves substantially better records than IMPALA across all the evaluation metrics. It also scores substantially better than all the methods in terms of mean HNS and median SABER scores. Though off-policy methods are known as privileged for HNS evaluation due to replay, CASA outperforms strong off-policy baseline Rainbow. Though LASER outperforms CASA in Median HNS and Mean SABER, CASA outperforms it in median SABER and mean HNS. Overall, the aforementioned results demonstrate the conflict-averse strategy efficiently boosts the performance in large-scale training scenarios and outperform strong on/off-policy algorithms. Hyperparameters and individual games are presented in Appendix F and Appendix G, respectively.

## 5 RELATED WORKS

Both value-based or policy-based approaches comply with the principle of GPI, but two GPI steps are coarsely related to each other such that jointly optimizing both functions might potentially bring conflicts. Despite of such crucial issue in GPI with function approximation, most decent model-free algorithms adopt a standard

policy improvement/evaluation regime without considering conflict diminishing properties. The issue of reducing conflicts among multiple models trained simultaneously was considered in earlier machine learning literature, such as for robust parameter estimation for multiple estimators under incomplete data (Robins & Rotnitzky, 1995; Lunceford & Davidian, 2004; Kang & Schafer, 2007) and multitask learning with gradient similarity measure (Chen et al., 2020; Yu et al., 2020; Javaloy & Valera, 2022).

When the idea is introduced to reinforcement learning, earliest attempts tackle conservative and safe policy iteration problems (Kakade & Langford, 2002; Hazan & Kale, 2011; Pirotta et al., 2013). Recently, more works have emerged to study GPI in a fine-grained manner. In (Ghosh et al., 2020), a new Bellman operator is introduced which implements GPI with a policy improvement operator and a projection operator, where the projection attempts to find the best approximation of policy among realizable policies. In (Raileanu & Fergus, 2021), the policy and value updates are decoupled by approximating two networks with representation regularization. In (Cobbe et al., 2021), GPI is separated into a policy improvement and a feature distillation step. On contrast to the aforementioned works, we tackle the conflicts in GPI at the gradient-level, with theoretical analysis. Our work is related to (Nachum et al., 2017), which utilizes both the unbiasedness and stability of on-policy training and the data efficiency of off-policy training to form a soft consistency error. Our work bridges the gap between the two GPI steps from an alternative angle of establishing a closer relationship between policy and value functions in their forms, without the focus on off-policy correction. Due to the difficulty of controlling the gap between GPI steps, we instead consider $\chi$. The condition $\chi = 1$ is close to compatible value function (Sutton et al., 1999; Kakade, 2001), shown in Section 3.1 and Appendix A.

## 6 LIMITATION

It's noticeable that CASA is only applied on discrete action space for now. We further find CASA applicable to any function approximation that is able to estimate advantage functions of all actions. We provide additional discussion on continuous action space in Appendix C.

Since $\pi$ shares all parameters of value functions, it brings $\chi = 1$ but sacrifices the *freedom* of $\pi$ to be parameterized by other parameters. We conjecture that CASA is one endpoint of a trade-off curve between $\chi$ and the *freedom* of $\pi$, where the other endpoint is that $\pi$ shares no parameter with value functions.

## 7 ETHICS AND REPRODUCIBILITY STATEMENT

This paper is aimed at academic issues in deep reinforcement learning, and the experiment used is also in the early stage, but it may provide opportunities for malicious applications of reinforcement learning in the future. We describe all details to reproduce the main experimental results in Appendix F.

## 8 CONCLUSION

This paper attempts to eliminate gradient inconsistency between policy improvement and policy evaluation. The proposed innovative actor-critic design **C**ritic **AS** an **A**ctor (CASA) enhances consistency of two GPI steps by satisfying a weaker compatible condition. We present both theoretical proof and empirical evaluation for CASA. The results show that our proposed method achieves state-of-the-art performance standards with noticeable performance gain over several strong baselines when evaluated on ALE 200 million (200M) benchmark. We also present several ablation studies, which demonstrates the effectiveness of the proposed method's theoretical properties. Future work includes studying the connection between the compatible condition and the gradient consistency between policy improvement and policy evalution.

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

## A   COMPATIBLE VALUE FUNCTION

The original policy gradient with compatible value function is stated as follow.

**Theorem 1** (Sutton et al. (1999)). *Let $Q_w$ be a state-action function with parameter $w$ and $\pi_\theta$ be a policy function with parameter $\theta$. If $Q_w$ satisfies $\mathbb{E}_\pi[(Q^\pi - Q_w)\nabla_w Q_w] = 0$ and $\nabla_w Q_w = \nabla_\theta \log \pi_\theta$, then*

$$\nabla_\theta \mathcal{J} = \mathbb{E}_\pi[Q_w \nabla_\theta \log \pi_\theta].$$

If we let $w = \theta$ in Theorem 1, where $Q_w$ and $\pi_\theta$ share parameters, we have the following theorem.

**Theorem 2.** *Let $Q_\theta$ be a state-action function with parameter $\theta$ and $\pi_\theta$ be a policy function with parameter $\theta$. If $Q_\theta$ satisfies $\mathbb{E}_\pi[(Q^\pi - Q_\theta)\nabla_\theta Q_\theta] = 0$ and $\nabla_\theta Q_\theta = \nabla_\theta \log \pi_\theta$, then*

$$\nabla_\theta \mathcal{J} = \mathbb{E}_\pi[Q_\theta \nabla_\theta \log \pi_\theta].$$

Define

$$\chi \stackrel{def}{=} \mathbb{E}_\pi[\cos < \nabla_\theta Q_\theta, \nabla_\theta \log \pi_\theta >].$$

We show that $\chi = 1$ is the necessary condition for the compatible condition $\nabla_\theta Q_\theta = \nabla_\theta \log \pi_\theta$.

**Theorem 3.** *i) If $\nabla_\theta Q_\theta \propto \nabla_\theta \log \pi_\theta$ for all states, then $\chi = 1$.*

*ii) If $\chi = 1$, then $\nabla_\theta Q_\theta \propto \nabla_\theta \log \pi_\theta$ for all states.*

By Theorem 3, $\chi = 1$ is equivalent to $\nabla_\theta Q_\theta \propto \nabla_\theta \log \pi_\theta$, and $\nabla_\theta Q_\theta \propto \nabla_\theta \log \pi_\theta$ is the necessary condition for $\nabla_\theta Q_\theta = \nabla_\theta \log \pi_\theta$, hence $\chi = 1$ is the necessary condition for $\nabla_\theta Q_\theta = \nabla_\theta \log \pi_\theta$.

*Proof.* i) Since $\nabla_\theta Q_\theta \propto \nabla_\theta \log \pi_\theta$, we have $< \nabla_\theta Q_\theta, \nabla_\theta \log \pi_\theta >= 0$. By definition of $\chi$, we have

$$\chi = \mathbb{E}_\pi[\cos < \nabla_\theta Q_\theta, \nabla_\theta \log \pi_\theta >] = \mathbb{E}_\pi[1] = 1.$$

ii) Since $\chi \leq 1$ and $\cos(x)$ is monotonic decreasing as $x$ goes from 0 to $\pi$, the equality $\chi = 1$ only holds when all states satisfy $\cos < \nabla_\theta Q_\theta, \nabla_\theta \log \pi_\theta >= 0$, which means $\nabla_\theta Q_\theta \propto \nabla_\theta \log \pi_\theta$.  □

## B  GRADIENTS BETWEEN POLICY IMPROVEMENT AND POLICY EVALUATION

|  | Function Approximation | Train Gradients | Cosine of Interested Angles |
|---|---|---|---|
| PPO | $(V, logit) = (V_\theta, logit_\theta)$ 
 $\pi = \text{softmax}(logit)$ | $0.5\nabla L_V + \nabla \mathcal{J}$ | |
| PPO ver.1 | $(Q, logit) = (Q_\theta, logit_\theta),$ 
 $\pi = \text{softmax}(logit)$ 
 $V = sg(\pi) \cdot Q$ | $0.5\nabla L_V + \nabla \mathcal{J}$ | $\cos < \nabla L_Q, \nabla \mathcal{J} >$ 
 $\cos < \nabla Q, \nabla \log \pi >$ |
| PPO ver.2 | $(Q, logit) = (Q_\theta, logit_\theta),$ 
 $pi = \text{softmax}(logit)$ 
 $V = sg(\pi) \cdot Q$ | $0.5\nabla L_V + \nabla L_Q + \nabla \mathcal{J}$ | $\cos < \nabla L_Q, \nabla \mathcal{J} >$ 
 $\cos < \nabla Q, \nabla \log \pi >$ |
| PPO+CASA | $(V, A) = (V_\theta, A_\theta),$ 
 $\pi = \text{softmax}(A/\tau),$ 
 $\bar{A} = A - sg(\pi) \cdot A$ 
 $Q = \bar{A} + sg(V)$ | $0.5\nabla L_V + \nabla L_Q + \nabla \mathcal{J}$ | $\cos < \nabla L_Q, \nabla \mathcal{J} >$ 
 $\cos < \nabla Q, \nabla \log \pi >$ |

Table 5: PPO is the original PPO. PPO ver.1 and PPO ver.2 are adapted versions to calculate $\nabla L_Q$. PPO+CASA is applying CASA on PPO, which is described in Sec. 4.2.

|  | Function Approximation | Train Gradients | Cosine of Interested Angles |
|---|---|---|---|
| R2D2 | $(V, A) = (V_\theta, A_\theta)$ 
 $Q = A + V$ 
 $\pi = \text{softmax}(A/\tau)$ | $\nabla L_Q$ | $\cos < \nabla L_Q, \nabla \mathcal{J} >$ |
| R2D2 ver.1 | $(V, A) = (V_\theta, A_\theta)$ 
 $Q = A + V$ 
 $\pi = \text{softmax}(A/\tau)$ | $0.5\nabla L_V + \nabla L_Q$ | $\cos < \nabla L_Q, \nabla \mathcal{J} >$ |
| R2D2+CASA | $(V, A) = (V_\theta, A_\theta),$ 
 $\pi = \text{softmax}(A/\tau),$ 
 $\bar{A} = A - sg(\pi) \cdot A$ 
 $Q = \bar{A} + sg(V)$ | $0.5\nabla L_V + \nabla L_Q + \nabla \mathcal{J}$ | $\cos < \nabla L_Q, \nabla \mathcal{J} >$ |

Table 6: R2D2 is the original R2D2. R2D2 ver.1 is adapted version to include $\nabla L_V$ for training. R2D2+CASA is applying CASA on R2D2, which is described in Sec. 4.2.

To understand the behavior of

$$\beta \stackrel{def}{=} < \mathbb{E}_\pi[(Q^\pi - Q_\theta)\nabla_\theta Q_\theta], \mathbb{E}_\pi[(Q^\pi - V_\theta)\nabla_\theta \log \pi_\theta] >$$

and

$$\chi \stackrel{def}{=} \mathbb{E}_\pi[\cos < \nabla_\theta Q_\theta, \nabla_\theta \log \pi_\theta >]$$

in reinforcement learning algorithms, we choose PPO as a representative for policy-based methods and R2D2 as a representative for value-based algorithms.

Define

$$L_V(\theta) = \mathbb{E}_\pi[(V^\pi - V_\theta)^2], \ L_Q(\theta) = \mathbb{E}_\pi[(Q^\pi - Q_\theta)^2],$$

and

$$\nabla_\theta \mathcal{J}(\theta) = \mathbb{E}_\pi\left[(Q^\pi - V_\theta)\nabla_\theta \log \pi\right].$$

We usually have above three kinds of loss functions in reinforcement learning, which aim to estimate the state values, state-action values and the policy. We do not talk about the estimations of $V^\pi$ and $Q^\pi$ as they are estimated as their usual way of PPO's and R2D2's. All hyperparameters are listed in Appendix F.

For brevity, we write

$$\cos < \nabla Q, \nabla \log \pi > = \mathbb{E}_\pi[\cos < \nabla_\theta Q_\theta, \nabla_\theta \log \pi_\theta >],$$

and

$$\cos < \nabla L_Q, \nabla \mathcal{J} > = \cos < \mathbb{E}_\pi[(Q^\pi - Q_\theta)\nabla_\theta Q_\theta], \mathbb{E}_\pi[(Q^\pi - V_\theta)\nabla_\theta \log \pi_\theta] >,$$
$$\cos < \nabla L_V, \nabla \mathcal{J} > = \cos < \mathbb{E}_\pi[(V^\pi - V_\theta)\nabla_\theta V_\theta], \mathbb{E}_\pi[(Q^\pi - V_\theta)\nabla_\theta \log \pi_\theta] >,$$
$$\cos < \nabla L_V, \nabla L_Q > = \cos < \mathbb{E}_\pi[(V^\pi - V_\theta)\nabla_\theta V_\theta], \mathbb{E}_\pi[(Q^\pi - Q_\theta)\nabla_\theta Q_\theta] > .$$

The fact that PPO only has $\nabla_\theta L_V$ and $\nabla_\theta \mathcal{J}$ and R2D2 only has $\nabla_\theta L_Q$ is the main difficulty to track $\cos(\beta)$ and $\chi$. To solve the problem, we adjust PPO and R2D2 with different versions.

For PPO, we displace the estimation of $V_\theta$ by $sg(\pi) \cdot Q_\theta$, where $Q_\theta$ is estimated by function approximation and $V_\theta$ is estimated by taking the expectation of $Q_\theta$. All versions of PPO are listed in Table 5.

For R2D2, we point out that though we apply $\epsilon$-greedy to interact with environments, $\epsilon$ is only used for exploration and the final target policy of value-based methods is simply $\arg\max Q_\theta$. Because $\arg\max Q_\theta$ breaks the gradient, we use a surrogate policy to approximate the gradient of policy improvement. Since R2D2 uses dueling structure and $\text{softmax}(A_\theta/\tau) = \text{softmax}(Q_\theta/\tau) \overset{\tau \to 0^+}{\longrightarrow} \arg\max Q_\theta$, we use $\pi_{surrogate} = \text{softmax}(A_\theta/\tau)$ to calculate the policy gradient. We only use $\pi_{surrogate}$ on learner to calculate the gradient, where the policy that interacts with environments is still $\epsilon$-greedy. All versions of R2D2 are listed in Table 6.

## C ON DISCUSSING APPLICATION OF CASA ON CONTINUOUS ACTION SPACE

As we can see CASA is only applied to discrete action space in the main context, we make a discussion on whether CASA is applicable on continuous action space. For brevity, we let $\tau = 1$ and write equation 6 as:

$$\begin{cases} \pi = \text{softmax}(A), \\ \bar{A} = A - \mathbb{E}_\pi[A], \\ Q = \bar{A} + sg(V). \end{cases} \tag{14}$$

The difficulty comes from estimating two quantities, one is $\text{softmax}(A)$, the other is $\mathbb{E}_\pi[A]$. This comes from the fact that discrete action space is countable so these two quantities are expressed in a closed-form, while continuous action space is uncountable so an accurate estimation of these two quantities is intractable. We can surely apply Monte Carlo methods to approximate, but a more elegant close-form expression may be preferred. Then this becomes another problem: *how to estimate (state-action values / advantages / policy probabilities) of all actions in a continuous action space efficiently without loss of generality?* This is another representational design problem, which is out of scope of this paper, so we don't touch much about it. But with the hope of inspiring a better solution to this problem, we provide one practical way of applying CASA on continuous action space based on kernel-based machine learning.

Let $a_0, \ldots, a_k$ to be basis actions in the action space. Let $A(s, a_0), \ldots, A(s, a_k)$ to be advantage functions for tuples of states and basis actions. They can either share parameters or be isolated. Let $K(\cdot, \cdot)$ be a kernel function defined on the product of two action spaces. For any $a$ in the action space, we can estimate $A(s, a)$ by a decomposition such like

$$A(s, a) = \frac{1}{Z_a}(K(a_0, a)A(s, a_0) + \cdots + K(a_k, a)A(s, a_k)),$$

where $Z_a = \sum_{i=0}^{k} K(a_i, a)$ is a normalization constant.

Since $K(\cdot, a)$ is a closed-form function of $a$, and $|\{A(s, a_0), \ldots, A(s, a_k)\}|$ is finite, we can make a closed-form expression of both $\text{softmax}(A)$ and $\mathbb{E}_\pi[A]$. Then we can apply CASA directly on this expression, with one function estimates $V$ and the other function estimates advantages of all actions in a closed-form with only state as input. The policy is defined directly by softmax of all advantages. In details, we define

$$\begin{cases} \pi(s, a) = \exp(A(s, a)) / \int_a \exp(A(s, a))da, \\ \bar{A}(s, a) = A(s, a) - \int_a sg(\pi(s, a))A(s, a)da, \\ Q(s, a) = \bar{A}(s, a) + sg(V(s)). \end{cases} \tag{15}$$

Then it satisfies the consistency of CASA on continuous action space.

$$\begin{aligned} \nabla \log \pi(s, a) &= \nabla A(s, a) - \frac{\nabla \int_a \exp(A(s, a))da}{\int_a \exp(A(s, a))da} \\ &= \nabla A(s, a) - \frac{\int_a \exp(A(s, a))\nabla A(s, a)da}{\int_a \exp(A(s, a))da} \\ &= \nabla A(s, a) - \int_a \frac{\exp(A(s, a))}{\int_a \exp(A(s, a))da}\nabla A(s, a)da \\ &= \nabla A(s, a) - \int_a \pi(s, a)\nabla A(s, a)da \\ &= \nabla \bar{A}(s, a) = \nabla Q(s, a). \end{aligned}$$

## D DR-TRACE

As CASA estimates $(V, Q, \pi)$, we would ask **i)** how to guarantee that $\tilde{\pi}_{VTrace} = \tilde{\pi}_{ReTrace}$, **ii)** how to exploit $(V, Q, \pi)$ to make a better estimation. Though we can apply V-Trace to estimate $V$ and ReTrace to estimate $Q$ with proper hyperparameters to guarantee $\tilde{\pi}_{VTrace} = \tilde{\pi}_{ReTrace}$, it's more reasonable to estimate $(V, Q)$ together. Inspired by Doubly Robust, which is shown to maximally reduce the variance, we introduce DR-Trace, which estimates $V$ by

$$V_{DR}^{\tilde{\pi}}(s_t) \stackrel{def}{=} \mathbb{E}_\mu[V(s_t) + \sum_{k \geq 0} \gamma^k c_{[t:t+k-1]} \rho_{t+k} \delta_{t+k}^{DR}],$$

where $\mu$ is the behavior policy, $\delta_t^{DR} \stackrel{def}{=} r_t + \gamma V(s_{t+1}) - Q(s_t, a_t)$ is one-step Doubly Robust error, $\rho_t \stackrel{def}{=} \min\{\frac{\pi_t}{\mu_t}, \bar{\rho}\}$ and $c_t \stackrel{def}{=} \min\{\frac{\pi_t}{\mu_t}, \bar{c}\}$ are clipped per-step importance sampling, $c_{[t:t+k]} \stackrel{def}{=} \prod_{i=0}^k c_{t+i}$.

With one step Bellman equation, we estimate $Q$ by

$$Q_{DR}^{\tilde{\pi}}(s_t, a_t) \stackrel{def}{=} \mathbb{E}_{s_{t+1}, r_t \sim p(\cdot, \cdot | s_t, a_t)}[r_t + \gamma V_{DR}^{\tilde{\pi}}(s_{t+1})]$$
$$= \mathbb{E}_\mu[Q(s_t, a_t) + \sum_{k \geq 0} \gamma^k c_{[t+1:t+k-1]} \tilde{\rho}_{t,k} \delta_{t+k}^{DR}],$$

where $\tilde{\rho}_{t,k} = 1_{\{k=0\}} + 1_{\{k>0\}} \rho_{t+k}$.

**Theorem 4.** *Define* $\bar{A} = A - \mathbb{E}_\pi[A]$, $Q = \bar{A} + sg(V)$,

$$\mathcal{T}(Q) \stackrel{def}{=} \mathbb{E}_\mu[Q(s_t, a_t) + \sum_{k \geq 0} \gamma^k c_{[t+1:t+k-1]} \tilde{\rho}_{t,k} \delta_{t+k}^{DR}],$$

$$\mathcal{S}(V) \stackrel{def}{=} \mathbb{E}_\mu[V(s_t) + \sum_{k \geq 0} \gamma^k c_{[t:t+k-1]} \rho_{t,k} \delta_{t+k}^{DR}],$$

$$\mathcal{U}(Q, V) = (\mathcal{T}(Q) - \mathbb{E}_\pi[Q] + \mathcal{S}(V), \mathcal{S}(V)),$$

$$\mathcal{U}^{(n)}(Q, V) = \mathcal{U}(\mathcal{U}^{(n-1)}(Q, V)),$$

*then* $\mathcal{U}^{(n)}(Q, V) \to (Q^{\tilde{\pi}}, V^{\tilde{\pi}})$ *that corresponds to*

$$\tilde{\pi}(a|s) = \frac{\min\{\bar{\rho}\mu(a|s), \pi(a|s)\}}{\sum_{b \in \mathcal{A}} \min\{\bar{\rho}\mu(b|s), \pi(b|s)\}}.$$

*as* $n \to +\infty$.

*Proof.* See Appendix E, Theorem E.1. $\qquad \square$

Theorem 4 shows that DR-Trace is a contraction mapping and $(V, Q)$ converges to $(V^{\tilde{\pi}}, Q^{\tilde{\pi}})$ that corresponds to

$$\tilde{\pi}(a|s) = \frac{\min\{\bar{\rho}\mu(a|s), \pi(a|s)\}}{\sum_{b \in \mathcal{A}} \min\{\bar{\rho}\mu(b|s), \pi(b|s)\}}.$$

According to our proof, DR-Trace should work similar to V-Trace and ReTrace, as the convergence rate and the limitation are same. We compare DR-Trace with V-Trace+ReTrace in Figure 5, where we replace estimation of state values by V-Trace and estimation of state-action values by ReTrace. We call V-Trace+ReTrace as No-DR-Trace for brevity. No-DR-Trace performs better on Breakout and ChopperCommand, but fails to make a breakthrough on Krull. Recalling the fact that Doubly Robust can maximally reduce the variance of Bellman error, No-DR-Trace is less stable but also potential to achieve a better performance. A conclusion cannot be made about No-DR-Trace, as this phenomenon means that No-DR-Trace is less stable than DR-Trace, but it also holds the potential to achieve a better performance.

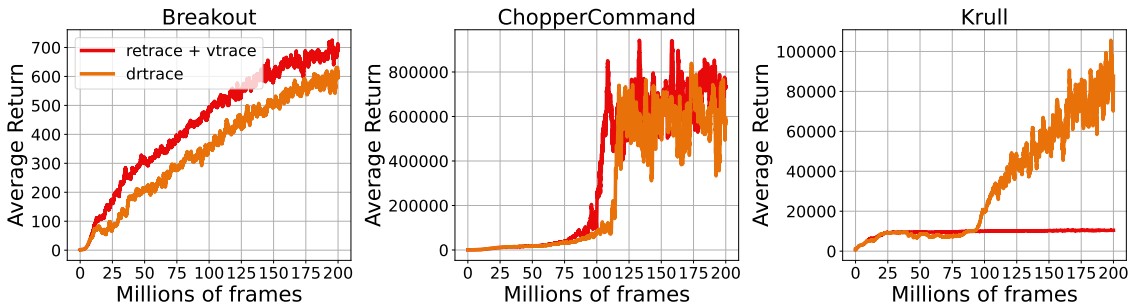

Figure 5: Ablation study for w/wo DR-Trace on Breakout, ChopperCommand and Krull.

## E   PROOFS

**Lemma E.1.** *(i) Define $\pi = softmax(A/\tau)$, then $\nabla \log \pi = (\mathbf{1} - \pi)\frac{\nabla A}{\tau}$. (ii) Denote sg to be stop gradient and define $\bar{A} = A - \mathbb{E}_\pi[A]$, $Q = \bar{A} + sg(V)$, then $\nabla Q = (\mathbf{1} - \pi)\nabla A$.*

*Proof.* As $Q = \bar{A} + sg(V) = A - sg(\pi) \cdot A + sg(V)$, it's obvious that $\nabla Q = (\mathbf{1} - \pi)\nabla A$.

For $\log \pi$, it's a standard derivative of cross entropy, so we have $\nabla \log \pi = (\mathbf{1} - \pi)\nabla(A/\tau) = (\mathbf{1} - \pi)\frac{\nabla A}{\tau}$.   $\square$

**Lemma E.2.** *Define $\bar{A} = A - \mathbb{E}_\pi[A]$, $Q = \bar{A} + sg(V)$, $\pi = softmax(A/\tau)$, then*

$$\mathbb{E}_\pi\left[(Q - V)\nabla \log \pi\right] = -\tau \nabla \boldsymbol{H}[\pi].$$

*Proof.* Since

$$\pi = \exp(A/\tau)/Z, \; Z = \int_{\mathcal{A}} \exp(A/\tau),$$

we have

$$A = \tau \log \pi + \tau \log Z.$$

Based on the observation that $\mathbb{E}_\pi\left[f(s)\nabla \log \pi(\cdot|s)\right] = 0$, we have

$$\mathbb{E}_\pi\left[\mathbb{E}_\pi[A] \cdot \nabla \log \pi\right] = 0,$$

$$\mathbb{E}_\pi\left[\log Z \cdot \nabla \log \pi\right] = 0.$$

On the one hand,

$$\begin{aligned}
\mathbb{E}_\pi\left[(Q - V)\nabla \log \pi\right] &= \mathbb{E}_\pi\left[A\nabla \log \pi\right] - \mathbb{E}_\pi\left[\mathbb{E}_\pi[A] \cdot \nabla \log \pi\right] \\
&= \tau\mathbb{E}_\pi\left[\log \pi\nabla \log \pi\right] + \tau\mathbb{E}_\pi\left[\log Z \cdot \nabla \log \pi\right] \\
&= \tau\mathbb{E}_\pi\left[\log \pi\nabla \log \pi\right].
\end{aligned}$$

On the other hand,

$$
\begin{aligned}
\nabla \mathbf{H}[\pi] &= -\nabla \int_{\mathcal{A}} \pi_i \log \pi_i \\
&= -\int_{\mathcal{A}} \nabla \pi_i \cdot \log \pi_i - \int_{\mathcal{A}} \pi_i \nabla \log \pi_i \\
&= -\int_{\mathcal{A}} \pi_i \nabla \log \pi_i \cdot \log \pi_i - \int_{\mathcal{A}} \pi_i \frac{\nabla \pi_i}{\pi_i} \\
&= -\mathbb{E}_\pi \left[ \log \pi \nabla \log \pi \right].
\end{aligned}
$$

Hence, $\mathbb{E}_\pi \left[ (Q - V) \nabla \log \pi \right] = -\tau \nabla \mathbf{H}[\pi]$. $\qquad \square$

**Theorem E.1.** *Define $\bar{A} = A - \mathbb{E}_\pi[A]$, $Q = \bar{A} + sg(V)$. Define*

$$
\mathscr{T}(Q) \overset{def}{=} \mathbb{E}_\mu[Q(s_t, a_t) + \sum_{k \geq 0} \gamma^k c_{[t+1:t+k-1]} \tilde{\rho}_{t,k} \delta_{t+k}^{DR}],
$$

$$
\mathscr{S}(V) \overset{def}{=} \mathbb{E}_\mu[V(s_t) + \sum_{k \geq 0} \gamma^k c_{[t:t+k-1]} \rho_{t,k} \delta_{t+k}^{DR}],
$$

$$
\mathscr{U}(Q, V) = (\mathscr{T}(Q) - \mathbb{E}_\pi[Q] + \mathscr{S}(V), \mathscr{S}(V)),
$$

$$
\mathscr{U}^{(n)}(Q, V) = \mathscr{U}(\mathscr{U}^{(n-1)}(Q, V)),
$$

*then $\mathscr{U}^{(n)}(Q, V) \to (Q^{\tilde{\pi}}, V^{\tilde{\pi}})$ that corresponds to*

$$
\tilde{\pi}(a|s) = \frac{\min \{\bar{\rho}\mu(a|s), \pi(a|s)\}}{\sum_{b \in \mathcal{A}} \min \{\bar{\rho}\mu(b|s), \pi(b|s)\}}.
$$

*as $n \to +\infty$.*

**Remark.** $\mathscr{T}(Q) - \mathbb{E}_\pi[Q] + \mathscr{S}(V)$ is **exactly** how $Q$ is updated at training time. Since $Q = \bar{A} + sg(V)$, if we apply gradient ascent on $Q$ and $V$ in directions $\nabla L_Q(\theta)$ and $\nabla L_V(\theta)$ respectively, change of $Q$ comes from two aspects. One comes from $\nabla L_Q(\theta)$, which changes $A$, the other comes from $\nabla L_V(\theta)$, which changes $V$. Because the gradient of $V$ is stopped when estimating $Q$, the latter is captured by "minus old baseline, add new baseline", which is $-\mathbb{E}_\pi[Q] + \mathscr{S}(V)$ in Theorem E.1.

*Proof.* Define

$$
\widetilde{\mathscr{T}}(Q) = -\mathbb{E}_\pi[Q] + \mathscr{T}(Q),
$$

$$
\widetilde{\mathscr{U}}(Q, V) = (\widetilde{\mathscr{T}}(Q), \mathscr{S}(V)),
$$

$$
\widetilde{\mathscr{U}}^{(n)}(Q, V) = \widetilde{\mathscr{U}}(\widetilde{\mathscr{U}}^{(n-1)}(Q, V)).
$$

By Lemma E.3, $\widetilde{\mathscr{T}}^{(n)}(Q)$ converges to some $A^*$ as $n \to \infty$. This process will not influence the estimation of $V$ as the gradient of $V$ is stopped when estimating $Q$. According to the proof, $A^*$ does not depend on $V$. By Lemma E.4, $\mathscr{S}^{(n)}(V)$ converges to some $V^*$ as $n \to \infty$.
Hence, we have

$$
\widetilde{\mathscr{U}}^{(n)}(Q, V) \to (A^*, V^*) \ as \ n \to +\infty.
$$

By definition,

$$
\mathscr{U}(Q, V) = (\widetilde{\mathscr{T}}(Q) + \mathscr{S}(V), \mathscr{S}(V)),
$$

we can regard $\widetilde{\mathscr{T}}(Q) + \mathscr{S}(V)$ as $Q$ and regard $\mathscr{S}(V)$ as $V$, then

$$
\begin{aligned}
\mathscr{U}^{(2)}(Q,V) &= \mathscr{U}(\widetilde{\mathscr{T}}(Q) + \mathscr{S}(V), \mathscr{S}(V)) \\
&= (\mathscr{T}(\widetilde{\mathscr{T}}(Q) + \mathscr{S}(V)) - \mathscr{S}(V) + \mathscr{S}^{(2)}(V), \mathscr{S}^{(2)}(V)) \\
&= (\widetilde{\mathscr{T}}^{(2)}(Q) + \mathscr{S}^{(2)}(V), \mathscr{S}^{(2)}(V)).
\end{aligned}
$$

By induction,

$$
\begin{aligned}
\mathscr{U}^{(n)}(Q,V) &= (\widetilde{\mathscr{T}}^{(n)}(Q) + \mathscr{S}^{(n)}(V), \mathscr{S}^{(n)}(V)) \\
&\to (A^* + V^*, V^*) \ as \ n \to +\infty.
\end{aligned}
$$

Same as (Espeholt et al., 2018),

$$
\tilde{\pi}(a|s) = \frac{\min\{\bar{\rho}\mu(a|s), \pi(a|s)\}}{\sum_{b \in \mathcal{A}} \min\{\bar{\rho}\mu(b|s), \pi(b|s)\}}.
$$

is the policy s.t. the Bellman equation holds, which is

$$
\mathbb{E}_\mu[\rho_t(r_t + \gamma V_{t+1} - V_t)|\mathscr{F}_t] = 0,
$$

and $\mathscr{U}(Q^{\tilde{\pi}}, V^{\tilde{\pi}}) = (Q^{\tilde{\pi}}, V^{\tilde{\pi}})$.
So we have $(A^* + V^*, V^*) = (Q^{\tilde{\pi}}, V^{\tilde{\pi}})$. □

**Lemma E.3.** *Define $\bar{A} = A - \mathbb{E}_\pi[A]$, $Q = \bar{A} + sg(V)$, then operator*

$$
\mathscr{T}(Q) \overset{def}{=} \mathbb{E}_\mu[Q(s_t, a_t) + \sum_{k \geq 0} \gamma^k c_{[t+1:t+k-1]} \tilde{\rho}_{t,k} \delta^{DR}_{t+k}]
$$

*is a contraction mapping w.r.t. $Q$.*

**Remark.** Note that $\mathscr{T}(Q)$ is exactly equation D.

Since $Q = A + sg(V)$, the gradient of $V$ is stopped when estimating $Q$, updating $Q$ will not change $V$, which is equivalent to updating $A$. Without loss of generality, we assume $V$ is fixed as $V^*$ in the proof.

*Proof.* $\bar{A} = A - \mathbb{E}_\pi[A]$ shows $\mathbb{E}_\pi[\bar{A}] = 0$, which guarantees that no matter how we update $A$, we always have $\mathbb{E}_\pi[Q] = V^*$.

Based on above observations, define

$$
\widetilde{\mathscr{T}}(Q) \overset{def}{=} -\mathbb{E}_\pi[Q] + \mathscr{T}(Q).
$$

It's obvious that we only need to prove $\widetilde{\mathscr{T}}(Q)$ is a contraction mapping.

For brevity, we denote

$$
Q_t = Q(s_t, a_t), A_t = A(s_t, a_t), V_t^* = V^*(s_t).
$$

Noticing that $\tilde{\rho}_{t,0} = 1$, let $\mathscr{F}$ represent filtration, we can rewrite $\widetilde{\mathscr{T}}$ as

$$
\begin{aligned}
\widetilde{\mathscr{T}}(Q) &= \mathbb{E}_\mu[A_t + \sum_{k \geq 0} \gamma^k c_{[t+1:t+k-1]} \tilde{\rho}_{t,k} \delta^{DR}_{t+k}] \\
&= \mathbb{E}_\mu[-V_t^* + \sum_{k \geq 0} \gamma^k c_{[t+1:t+k-1]} \tilde{\rho}_{t,k} r_{t+k} + \sum_{k \geq 0} \gamma^{k+1} c_{[t+1:t+k-1]} \Delta_k],
\end{aligned}
\tag{16}
$$

where

$$\Delta_k = \mathbb{E}_\mu \left[ \tilde{\rho}_{t,k} V^*_{t+k+1} - c_{t+k} \tilde{\rho}_{t,k+1} Q_{t+k+1} | \mathscr{F}_{t+k} \right]. \tag{17}$$

By definition of $Q$,

$$\mathbb{E}_\mu[V^*_{t+k+1} | \mathscr{F}_{t+k}] = \mathbb{E}_\mu[\mathbb{E}_\pi[Q_{t+k+1} | \mathscr{F}_{t+k+1}] | \mathscr{F}_{t+k}],$$

we can rewrite equation 17 as

$$\Delta_k = \mathbb{E}_\mu[(\tilde{\rho}_{t,k} \frac{\pi_{t+k+1}}{\mu_{t+k+1}} - c_{t+k} \tilde{\rho}_{t,k+1}) Q_{t+k+1} | \mathscr{F}_{t+k}]. \tag{18}$$

For any $Q_1 = A_1 + sg(V^*)$, $Q_2 = A_2 + sg(V^*)$, since

$$\mathbb{E}_\mu[(\tilde{\rho}_{t,k} \frac{\pi_{t+k+1}}{\mu_{t+k+1}} - c_{t+k} \tilde{\rho}_{t,k+1}) | \mathscr{F}_{t+k}] \geq 0,$$

by equation 16 equation 18, we have

$$||\widetilde{\mathscr{T}}(Q_1) - \widetilde{\mathscr{T}}(Q_2)|| \leq \mathcal{C}||Q_1 - Q_2||,$$

where

$$\mathcal{C} = \mathbb{E}_\mu[\sum_{k\geq 0} \gamma^{k+1} c_{[t+1:t+k-1]} (\tilde{\rho}_{t,k} \frac{\pi_{t+k+1}}{\mu_{t+k+1}} - c_{t+k} \tilde{\rho}_{t,k+1})]$$

$$= \mathbb{E}_\mu[1 - 1 + \sum_{k\geq 0} \gamma^{k+1} c_{[t+1:t+k-1]} (\tilde{\rho}_{t,k} - c_{t+k} \tilde{\rho}_{t,k+1})]$$

$$= 1 - (1-\gamma) \mathbb{E}_\mu[\sum_{k\geq 0} \gamma^k c_{[t+1:t+k-1]} \tilde{\rho}_{t,k}]$$

$$\leq 1 - (1-\gamma) < 1.$$

Hence, $\widetilde{\mathscr{T}}(Q)$ is a contraction mapping and converges to some fixed function, which we denote as $A^*$. So $\mathscr{T}(Q)$ is also a contraction mapping and converges to $A^* + V^*$. □

**Lemma E.4.** *Define $Q = A + sg(V)$ with $\mathbb{E}_\pi[A] = 0$, then operator*

$$\mathscr{S}(V) \stackrel{def}{=} \mathbb{E}_\mu[V(s_t) + \sum_{k\geq 0} \gamma^k c_{[t:t+k-1]} \rho_{t,k} \delta^{DR}_{t+k}]$$

*is a contraction mapping w.r.t. $V$.*

**Remark.** Note that $\mathscr{S}(V)$ is exactly equation D.

*Proof.* Same as Lemma E.3, we can get

$$\Delta_k = \mathbb{E}_\mu \left[ (\rho_{t+k} - c_{t+k} \rho_{t+k+1}) V_{t+k+1} - c_{t+k} \rho_{t+k+1} A^*_{t+k+1} | \mathscr{F}_{t+k} \right],$$

so we have

$$\Delta_k^1 - \Delta_k^2 = \mathbb{E}_\mu \left[ (\rho_{t+k} - c_{t+k} \rho_{t+k+1}) \cdot (V^1_{t+k+1} - V^2_{t+k+1}) | \mathscr{F}_{t+k} \right].$$

The remaining proof is identical to (Espeholt et al., 2018)'s. □

## F HYPERPARAMETERS

Our python packages are shown in Table 7.

| Package | Version |
|---|---|
| ale-py | 0.6.0.dev20200207 |
| gym | 0.19.0 |
| tensorflow | 1.15.2 |
| opencv-python | 4.1.2.30 |
| opencv-contrib-python | 4.4.0.46 |

Table 7: Versions for python packages among all experiments.

All experiments follow the shared hyperparameters as in Table 8. The specific hyperparameters for PPO, R2D2 and CASA+DR-Trace are shown in Table 9, Table 10 and Table 11. The only exceptions are $V$-loss scaling, $Q$-loss scaling and $\pi$-loss scaling, which may be zero depending on some specific ablation settings. We will state these three hyperparameters every time in all experiments.

| Parameter | Value |
|---|---|
| Atari Version | NoFrameskip-v4 |
| Atari Wrapper | gym.wrappers.atari_preprocessing |
| Image Size | (84, 84) |
| Grayscale | Yes |
| Num. Action Repeats | 4 |
| Num. Frame Stacks | 4 |
| Action Space | Full |
| End of Episode When Life Lost | No |
| Num. Environments | 160 |
| Random No-ops | 30 |
| Burn-in Stored Recurrent State | Yes |
| Bootstrap | Yes |
| Optimizer | Adam Weight Decay |
| Weight Decay Rate | 0.01 |
| Weight Decay Schedule | Anneal linearly to 0 |
| Learning Rate | 5e-4 |
| Warmup Steps | 4000 |
| Learning Rate Schedule | Anneal linearly to 0 |
| AdamW $\beta_1$ | 0.9 |
| AdamW $\beta_2$ | 0.98 |
| AdamW $\epsilon$ | 1e-6 |
| AdamW Clip Norm | 50.0 |
| Learner Push Model Every $n$ Steps | 25 |
| Actor Pull Model Every $n$ Steps | 64 |

Table 8: Configurations for shared hyperparameters among all experiments.

| Parameter | Value |
|---|---|
| Num. States | 50M |
| Sample Reuse | 1 |
| Reward Shape | $\text{clip}(r, 0, 1)$ |
| Burn-in | 0 |
| Seq-length | 40 |
| Discount ($\gamma$) | 0.995 |
| Batch size | 8 |
| Backbone | IMPALA,shallow without LSTM |
| PPO clip $\epsilon$ | 0.2 |
| GAE $\lambda$ | 0.8 |
| Temperature ($\tau$) | 0.1 |

Table 9: Hyperparameter configurations for PPO.

| Parameter | Value |
|---|---|
| Num. States | 50M |
| Sample Reuse | 2 |
| Target Shape | $Q_t^{\tilde{\pi}} = h(\sum_{i=0}^{n-1} \gamma^i r_{t+i} + \gamma^n h^{-1}(\text{Double}(Q_{t+n})))$ |
| Target Shape Function $h$ | $h(x) = \text{sign}(x) \cdot (\sqrt{|x|+1} - 1) + 10^{-3}x$ |
| Bootstrap Length $n$ | 5 |
| $\epsilon$-greedy | $\epsilon \sim 0.4^{\text{uniform}(1,8)}$ |
| PER Sample Temperature $\alpha$ | 0.9 |
| PER Buffer Size | 400000 |
| Burn-in | 0 |
| Seq-length | 40 |
| Discount ($\gamma$) | 0.997 |
| Batch size | 8 |
| Backbone | IMPALA,shallow without LSTM |
| Temperature ($\tau$) | 0.1 |

Table 10: Hyperparameter configurations for R2D2.

| Parameter | Value |
|---|---|
| Num. States | 200M |
| Sample Reuse | 2 |
| Reward Shape | $\log(|r| + 1.0) \cdot (2 \cdot 1_{\{r \geq 0\}} - 1_{\{r < 0\}})$ |
| Burn-in | 40 |
| Seq-length | 80 |
| Discount ($\gamma$) | 0.997 |
| Batch size | 64 |
| Backbone | IMPALA,deep |
| LSTM Units | 256 |
| $V$-loss Scaling ($\alpha_1$) | 1.0 |
| $Q$-loss Scaling ($\alpha_2$) | 10.0 |
| $\pi$-loss Scaling ($\alpha_3$) | 10.0 |
| Temperature ($\tau$) | 1.0 |
| Importance Sampling Clip $\bar{c}$ | 1.05 |
| Importance Sampling Clip $\bar{\rho}$ | 1.05 |

Table 11: Hyperparameter configurations for CASA + DR-Trace.

## G    EVALUATION OF CASA ON ATARI GAMES

Random scores and average human's scores are from (Badia et al., 2020). Human World Records (HWR) are from (Toromanoff et al., 2019). Rainbow's scores are from (Hessel et al., 2017). IMPALA's scores are from (Espeholt et al., 2018). LASER's scores are from (Schmitt et al., 2020), no sweep at 200M.

| Games | RND | HUMAN | RAINBOW | HNS(%) | IMPALA | HNS(%) | LASER | HNS(%) | CASA | HNS(%) |
|---|---|---|---|---|---|---|---|---|---|---|
| Scale | | | 200M | | 200M | | 200M | | 200M | |
| alien | 227.8 | 7127.8 | 9491.7 | 134.26 | 15962.1 | 228.03 | **35565.9** | **512.15** | 26137 | 375.50 |
| amidar | 5.8 | 1719.5 | **5131.2** | **299.08** | 1554.79 | 90.39 | 1829.2 | 106.4 | 560 | 32.34 |
| assault | 222.4 | 742 | 14198.5 | 2689.78 | 19148.47 | 3642.43 | 21560.4 | 4106.62 | 16228 | 3080.37 |
| asterix | 210 | 8503.3 | **428200** | **5160.67** | 300732 | 3623.67 | 240090 | 2892.46 | 213580 | 2572.80 |
| asteroids | 719 | 47388.7 | 2712.8 | 4.27 | 108590.05 | 231.14 | **213025** | **454.91** | 80339 | 170.60 |
| atlantis | 12850 | 29028.1 | 826660 | 5030.32 | 849967.5 | 5174.39 | 841200 | 5120.19 | **3211600** | **19772.10** |
| bank heist | 14.2 | 753.1 | **1358** | **181.86** | 1223.15 | 163.61 | 569.4 | 75.14 | 895.3 | 119.24 |
| battle zone | 236 | 37187.5 | 62010 | 167.18 | 20885 | 55.88 | 64953.3 | 175.14 | **91269** | **246.36** |
| beam rider | 363.9 | 16926.5 | 16850.2 | 99.54 | 32463.47 | 193.81 | **90881.6** | **546.52** | 57456 | 344.70 |
| berzerk | 123.7 | 2630.4 | 2545.6 | 96.62 | 1852.7 | 68.98 | **25579.5** | **1015.51** | 1648 | 60.81 |
| bowling | 23.1 | 160.7 | 30 | 5.01 | 59.92 | 26.76 | 48.3 | 18.31 | **162.4** | **101.24** |
| boxing | 0.1 | 12.1 | 99.6 | 829.17 | 99.96 | 832.17 | **100** | **832.5** | 98.3 | 818.33 |
| breakout | 1.7 | 30.5 | 417.5 | 1443.75 | **787.34** | **2727.92** | 747.9 | 2590.97 | 624.3 | 2161.81 |
| centipede | 2090.9 | 12017 | 8167.3 | 61.22 | 11049.75 | 90.26 | **292792** | **2928.65** | 102600 | 1012.57 |
| chopper command | 811 | 7387.8 | 16654 | 240.89 | 28255 | 417.29 | **761699** | **11569.27** | 616690 | 9364.42 |
| crazy climber | 10780.5 | 36829.4 | **168788.5** | **630.80** | 136950 | 503.69 | 167820 | 626.93 | 161250 | 600.70 |
| defender | 2874.5 | 18688.9 | 55105 | 330.27 | 185203 | 1152.93 | 336953 | 2112.50 | **421600** | **2647.75** |
| demon attack | 152.1 | 1971 | 111185 | 6104.40 | 132826.98 | 7294.24 | 133530 | 7332.89 | **291590** | **16022.76** |
| double dunk | -18.6 | -16.4 | -0.3 | 831.82 | -0.33 | 830.45 | 14 | 1481.82 | **20.25** | **1765.91** |
| enduro | 0 | 860.5 | 2125.9 | 247.05 | 0 | 0.00 | 0 | 0.00 | **10019** | **1164.32** |
| fishing derby | -91.7 | -38.8 | 31.3 | 232.51 | 44.85 | 258.13 | 45.2 | 258.79 | **53.24** | **273.99** |
| freeway | 0 | 29.6 | **34** | **114.86** | 0 | 0.00 | 0 | 0.00 | 3.46 | 11.69 |
| frostbite | 65.2 | 4334.7 | **9590.5** | **223.10** | 317.75 | 5.92 | 5083.5 | 117.54 | 1583 | 35.55 |
| gopher | 257.6 | 2412.5 | 70354.6 | 3252.91 | 66782.3 | 3087.14 | 114820.7 | 5316.40 | **188680** | **8743.90** |
| gravitar | 173 | 3351.4 | 1419.3 | 39.21 | 359.5 | 5.87 | 1106.2 | 29.36 | **4311** | **130.19** |
| hero | 1027 | 30826.4 | **55887.4** | **184.10** | 33730.55 | 109.75 | 31628.7 | 102.69 | 24236 | 77.88 |
| ice hockey | -11.2 | 0.9 | 1.1 | 101.65 | 3.48 | 121.32 | **17.4** | **236.36** | 1.56 | 105.45 |
| jamesbond | 29 | 302.8 | 19809 | 72.24 | 601.5 | 209.09 | **37999.8** | **13868.08** | 12468 | 4543.10 |
| kangaroo | 52 | 3035 | **14637.5** | **488.05** | 1632 | 52.97 | 14308 | 477.91 | 5399 | 179.25 |
| krull | 1598 | 2665.5 | 8741.5 | 669.18 | 8147.4 | 613.53 | 9387.5 | 729.70 | **64347** | **5878.13** |
| kung fu master | 258.5 | 22736.3 | 52181 | 230.99 | 43375.5 | 191.82 | **607443** | **2701.26** | 124630.1 | 553.31 |
| montezuma revenge | 0 | **4753.3** | 384 | 8.08 | 0 | 0.00 | 0.3 | 0.01 | 2488.4 | 52.35 |
| ms pacman | 307.3 | 6951.6 | 5380.4 | 76.35 | 7342.32 | 105.88 | 6565.5 | 94.19 | **7579** | **109.44** |
| name this game | 2292.3 | 8049 | 13136 | 188.37 | 21537.2 | 334.30 | 26219.5 | 415.64 | **32098** | **517.76** |
| phoenix | 761.5 | 7242.6 | 108529 | 1662.80 | 210996.45 | 3243.82 | **519304** | **8000.84** | 498590 | 7681.23 |
| pitfall | -229.4 | **6463.7** | 0 | 3.43 | -1.66 | 3.40 | -0.6 | 3.42 | -17.8 | 3.16 |
| pong | -20.7 | 14.6 | 20.9 | 117.85 | 20.98 | 118.07 | **21** | **118.13** | 20.39 | 116.40 |
| private eye | 24.9 | **69571.3** | 4234 | 6.05 | 98.5 | 0.11 | 96.3 | 0.10 | 134.1 | 0.16 |
| qbert | 163.9 | 13455.0 | 33817.5 | 253.20 | **351200.12** | **2641.14** | 21449.6 | 160.15 | 27371 | 204.70 |
| riverraid | 1338.5 | 17118.0 | 22920.8 | 136.77 | 29608.05 | 179.15 | **40362.7** | **247.31** | 11182 | 62.38 |
| road runner | 11.5 | 7845 | 62041 | 791.85 | 57121 | 729.04 | 45289 | 578.00 | **251360** | **3208.64** |
| robotank | 2.2 | 11.9 | 61.4 | 610.31 | 12.96 | 110.93 | **62.1** | **617.53** | 10.44 | 84.95 |
| seaquest | 68.4 | 42054.7 | 15898.9 | 37.70 | 1753.2 | 4.01 | 2890.3 | 6.72 | 11862 | 28.09 |
| skiing | -17098 | **-4336.9** | -12957.8 | 32.44 | -10180.38 | 54.21 | -29968.4 | -100.86 | -12730 | 34.23 |
| solaris | 1236.3 | **12326.7** | 3560.3 | 20.96 | 2365 | 10.18 | 2273.5 | 9.35 | 2319 | 9.76 |
| space invaders | 148 | 1668.7 | 18789 | 1225.82 | 43595.78 | 2857.09 | **51037.4** | **3346.45** | 3031 | 189.58 |
| star gunner | 664 | 10250 | 127029 | 1318.22 | 200625 | 2085.97 | 321528 | 3347.21 | **337150** | **3510.18** |
| surround | -10 | 6.5 | **9.7** | **119.39** | 7.56 | 106.42 | 8.4 | 111.52 | -10 | 0.00 |
| tennis | -23.8 | -8.3 | 0 | 153.55 | 0.55 | 157.10 | **12.2** | **232.26** | -21.05 | 17.74 |
| time pilot | 3568 | 5229.2 | 12926 | 563.36 | 48481.5 | 2703.84 | 105316 | 6125.34 | 84341 | 4862.62 |
| tutankham | 11.4 | 167.6 | 241 | 146.99 | 292.11 | 179.71 | 278.9 | 171.25 | **381** | **236.62** |
| up n down | 533.4 | 11693.2 | 125755 | 1122.08 | 332546.75 | 2975.08 | 345727 | 3093.19 | 416020 | 3723.06 |
| venture | 0 | **1187.5** | 5.5 | 0.46 | 0 | 0.00 | 0 | 0.00 | 0 | 0.00 |
| video pinball | 0 | 17667.9 | 533936.5 | 3022.07 | **572898.27** | **3242.59** | 511835 | 2896.98 | 297920 | 1686.22 |
| wizard of wor | 563.5 | 4756.5 | 17862.5 | 412.57 | 9157.5 | 204.96 | **29059.3** | **679.60** | 26008 | 606.83 |
| yars revenge | 3092.9 | 54576.9 | 102557 | 193.19 | 84231.14 | 157.60 | **166292.3** | **316.99** | 118730 | 224.61 |
| zaxxon | 32.5 | 9173.3 | 22209.5 | 242.62 | 32935.5 | 359.96 | 41118 | 449.47 | **46070.8** | **503.66** |
| MEAN HNS(%) | 0.00 | 100.00 | | 873.97 | | 957.34 | | 1741.36 | | 1941.08 |
| MEDIAN HNS(%) | 0.00 | 100.00 | | 230.99 | | 191.82 | | 454.91 | | 246.36 |

| Games | RND | HWR | RAINBOW | SABER(%) | IMPALA | SABER(%) | LASER | SABER(%) | CASA | SABER(%) |
|---|---|---|---|---|---|---|---|---|---|---|
| Scale | | | 200M | | 200M | | 200M | | 200M | |
| alien | 227.8 | **251916** | 9491.7 | 3.68 | 15962.1 | 6.25 | 976.51 | 14.04 | 26137 | 10.29 |
| amidar | 5.8 | **104159** | 5131.2 | 4.92 | 1554.79 | 1.49 | 1829.2 | 1.75 | 560 | 0.53 |
| assault | 222.4 | 8647 | 14198.5 | 165.90 | 19148.47 | 200.00 | **21560.4** | **200.00** | 16228 | 189.99 |
| asterix | 210 | **1000000** | 428200 | 42.81 | 300732 | 30.06 | 240090 | 23.99 | 213580 | 21.34 |
| asteroids | 719 | **10506650** | 2712.8 | 0.02 | 108590.05 | 1.03 | 213025 | 2.02 | 80339 | 0.76 |
| atlantis | 12850 | **10604840** | 826660 | 7.68 | 849967.5 | 7.90 | 841200 | 7.82 | 3211600 | 30.20 |
| bank heist | 14.2 | **82058** | 1358 | 1.64 | 1223.15 | 1.47 | 569.4 | 0.68 | 895.3 | 1.07 |
| battle zone | 236 | **801000** | 62010 | 7.71 | 20885 | 2.58 | 64953.3 | 8.08 | 91269 | 11.37 |
| beam rider | 363.9 | **999999** | 16850.2 | 1.65 | 32463.47 | 3.21 | 90881.6 | 9.06 | 57456 | 5.71 |
| berzerk | 123.7 | **1057940** | 2545.6 | 0.23 | 1852.7 | 0.16 | 25579.5 | 2.41 | 1648 | 0.14 |
| bowling | 23.1 | **300** | 30 | 2.49 | 59.92 | 13.30 | 48.3 | 9.10 | 162.4 | 50.31 |
| boxing | 0.1 | **100** | 99.6 | 99.60 | 99.96 | 99.96 | 100 | **100.00** | 98.3 | 98.3 |
| breakout | 1.7 | **864** | 417.5 | 48.22 | 787.34 | 91.11 | 747.9 | 86.54 | 624.3 | 72.20 |
| centipede | 2090.9 | **1301709** | 8167.3 | 0.47 | 11049.75 | 0.69 | 292792 | 22.37 | 102600 | 7.73 |
| chopper command | 811 | **999999** | 16654 | 1.59 | 28255 | 2.75 | 761699 | 76.15 | 616690 | 61.64 |
| crazy climber | 10780.5 | **219900** | 168788.5 | 75.56 | 136950 | 60.33 | 167820 | 75.10 | 161250 | 71.95 |
| defender | 2874.5 | **6010500** | 55105 | 0.87 | 185203 | 3.03 | 336953 | 5.56 | 421600 | 6.97 |
| demon attack | 152.1 | **1556345** | 111185 | 7.13 | 132826.98 | 8.53 | 133530 | 8.57 | 291590 | 18.73 |
| double dunk | -18.6 | **21** | -0.3 | 46.21 | -0.33 | 46.14 | 14 | 82.32 | 20.25 | 98.11 |
| enduro | 0 | 9500 | 2125.9 | 22.38 | 0 | 0.00 | 0 | 0.00 | **10019** | **105.46** |
| fishing derby | -91.7 | **71** | 31.3 | 75.60 | 44.85 | 83.93 | 45.2 | 84.14 | 53.24 | 89.08 |
| freeway | 0 | **38** | 34 | 89.47 | 0 | 0.00 | 0 | 0.00 | 3.46 | 9.11 |
| frostbite | 65.2 | **454830** | 9590.5 | 2.09 | 317.75 | 0.06 | 5083.5 | 1.10 | 1583 | 0.33 |
| gopher | 257.6 | **355060** | 70354.6 | 19.76 | 66782.3 | 18.75 | 114820.7 | 32.29 | 188680 | 53.11 |
| gravitar | 173 | **162850** | 1419.3 | 0.77 | 359.5 | 0.11 | 1106.2 | 0.57 | 4311 | 2.54 |
| hero | 1027 | **1000000** | 55887.4 | 5.49 | 33730.55 | 3.27 | 31628.7 | 3.06 | 24236 | 2.32 |
| ice hockey | -11.2 | **36** | 1.1 | 26.06 | 3.48 | 31.10 | 17.4 | 60.59 | 1.56 | 27.03 |
| jamesbond | 29 | **45550** | 19809 | 43.45 | 601.5 | 1.26 | 37999.8 | 83.41 | 12468 | 27.33 |
| kangaroo | 52 | **1424600** | 14637.5 | 1.02 | 1632 | 0.11 | 14308 | 1.00 | 5399 | 0.38 |
| krull | 1598 | **104100** | 8741.5 | 6.97 | 8147.4 | 6.39 | 9387.5 | 7.60 | 64347 | 61.22 |
| kung fu master | 258.5 | **1000000** | 52181 | 5.19 | 43375.5 | 4.31 | 607443 | 60.73 | 124630.1 | 12.44 |
| montezuma revenge | 0 | **1219200** | 384 | 0.03 | 0 | 0.00 | 0.3 | 0.00 | 2488.4 | 0.20 |
| ms pacman | 307.3 | **290090** | 5380.4 | 1.75 | 7342.32 | 2.43 | 6565.5 | 2.16 | 7579 | 2.51 |
| name this game | 2292.3 | 25220 | 13136 | 47.30 | 21537.2 | 83.94 | 26219.5 | 104.36 | **32098** | **130.00** |
| phoenix | 761.5 | **4014440** | 108529 | 2.69 | 210996.45 | 5.24 | 519304 | 12.92 | 498590 | 12.40 |
| pitfall | -229.4 | **114000** | 0 | 0.20 | -1.66 | 0.20 | -0.6 | 0.20 | -17.8 | 0.19 |
| pong | -20.7 | **21** | 20.9 | 99.76 | 20.98 | 99.95 | 21 | **100.00** | 20.39 | 98.54 |
| private eye | 24.9 | **101800** | 4234 | 4.14 | 98.5 | 0.07 | 96.3 | 0.07 | 134.1 | 0.11 |
| qbert | 163.9 | **2400000** | 33817.5 | 1.40 | 351200.12 | 14.63 | 21449.6 | 0.89 | 27371 | 1.13 |
| riverraid | 1338.5 | **1000000** | 22920.8 | 2.16 | 29608.05 | 2.83 | 40362.7 | 3.91 | 11182 | 0.99 |
| road runner | 11.5 | **2038100** | 62041 | 3.04 | 57121 | 2.80 | 45289 | 2.22 | 251360 | 12.33 |
| robotank | 2.2 | **76** | 61.4 | 80.22 | 12.96 | 14.58 | 62.1 | 81.17 | 10.44 | 11.17 |
| seaquest | 68.4 | **999999** | 15898.9 | 1.58 | 1753.2 | 0.17 | 2890.3 | 0.28 | 11862 | 1.18 |
| skiing | -17098 | **-3272** | -12957.8 | 29.95 | -10180.38 | 50.03 | -29968.4 | -93.09 | -12730 | 31.59 |
| solaris | 1236.3 | **111420** | 3560.3 | 2.11 | 2365 | 1.02 | 2273.5 | 0.94 | 2319 | 0.98 |
| space invaders | 148 | **621535** | 18789 | 3.00 | 43595.78 | 6.99 | 51037.4 | 8.19 | 3031 | 0.46 |
| star gunner | 664 | 77400 | 127029 | 164.67 | 200625 | 200.00 | 321528 | 200.00 | **337150** | **200.00** |
| surround | -10 | 9.6 | **9.7** | **100.51** | 7.56 | 89.59 | 8.4 | 93.88 | -10 | 0.00 |
| tennis | -23.8 | **21** | 0 | 53.13 | 0.55 | 54.35 | 12.2 | 80.36 | -21.05 | 6.14 |
| time pilot | 3568 | 65300 | 12926 | 15.16 | 48481.5 | 72.76 | **105316** | **164.82** | 84341 | 130.84 |
| tutankham | 11.4 | **5384** | 241 | 4.27 | 292.11 | 5.22 | 278.9 | 4.98 | 381 | 6.88 |
| up n down | 533.4 | 82840 | 125755 | 152.14 | 332546.75 | 200.00 | 345727 | 200.00 | **416020** | **200.00** |
| venture | 0 | **38900** | 5.5 | 0.01 | 0 | 0.00 | 0 | 0.00 | 0 | 0.00 |
| video pinball | 0 | **89218328** | 533936.5 | 0.60 | 572898.27 | 0.64 | 511835 | 0.57 | 297920 | 0.33 |
| wizard of wor | 563.5 | **395300** | 17862.5 | 4.38 | 9157.5 | 2.18 | 29059.3 | 7.22 | 26008 | 6.45 |
| yars revenge | 3092.9 | **15000105** | 102557 | 0.66 | 84231.14 | 0.54 | 166292.3 | 1.09 | 118730 | 0.77 |
| zaxxon | 32.5 | **83700** | 22209.5 | 26.51 | 32935.5 | 39.33 | 41118 | 49.11 | 46070.8 | 55.03 |
| MEAN SABER(%) | 0.00 | 100.00 | | 28.39 | | 29.45 | | 36.78 | | 36.10 |
| MEDIAN SABER(%) | 0.00 | 100.00 | | 4.92 | | 4.31 | | 8.08 | | 10.29 |

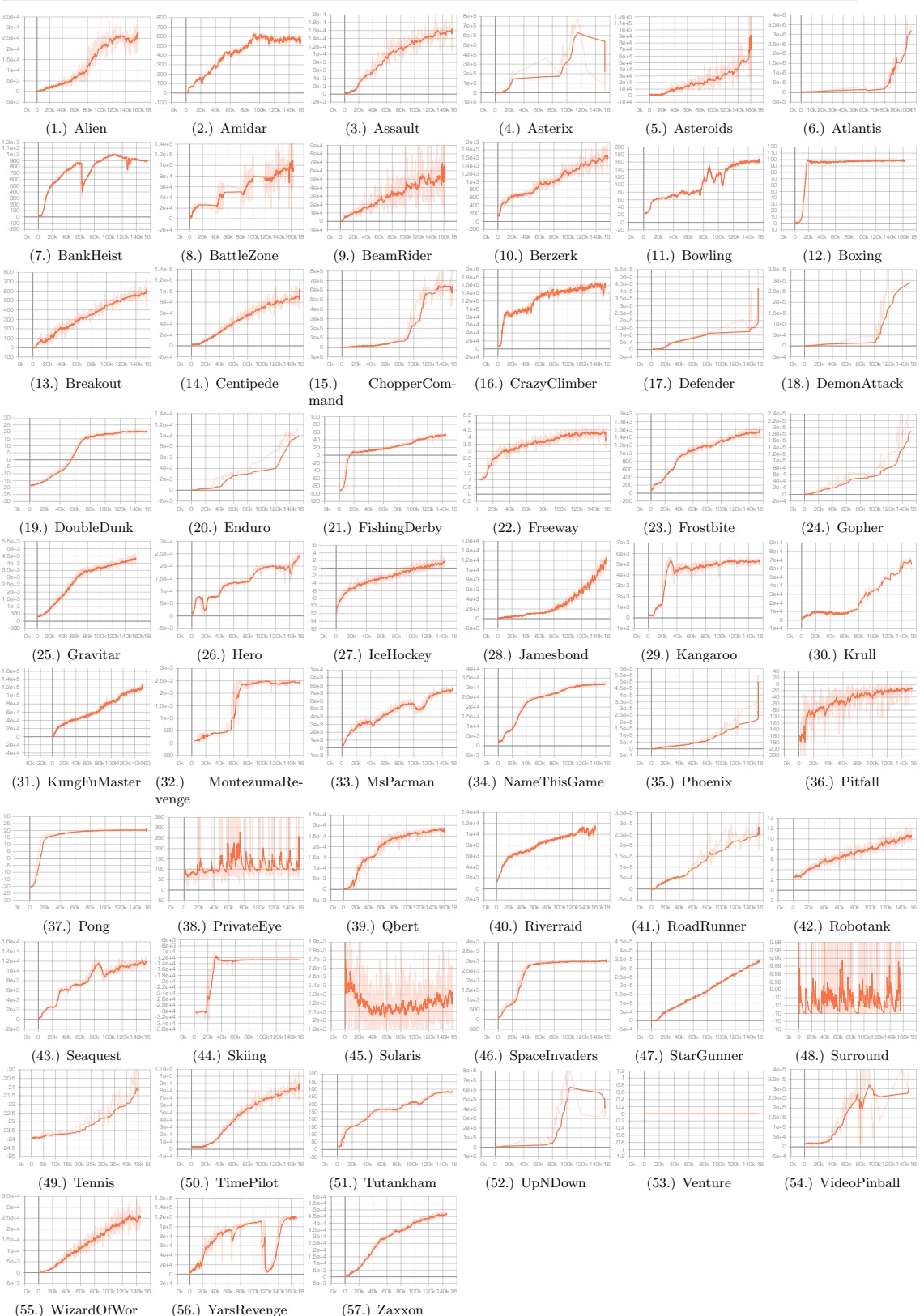

(1.) Alien  (2.) Amidar  (3.) Assault  (4.) Asterix  (5.) Asteroids  (6.) Atlantis

(7.) BankHeist  (8.) BattleZone  (9.) BeamRider  (10.) Berzerk  (11.) Bowling  (12.) Boxing

(13.) Breakout  (14.) Centipede  (15.) ChopperCommand  (16.) CrazyClimber  (17.) Defender  (18.) DemonAttack

(19.) DoubleDunk  (20.) Enduro  (21.) FishingDerby  (22.) Freeway  (23.) Frostbite  (24.) Gopher

(25.) Gravitar  (26.) Hero  (27.) IceHockey  (28.) Jamesbond  (29.) Kangaroo  (30.) Krull

(31.) KungFuMaster  (32.) MontezumaRevenge  (33.) MsPacman  (34.) NameThisGame  (35.) Phoenix  (36.) Pitfall

(37.) Pong  (38.) PrivateEye  (39.) Qbert  (40.) Riverraid  (41.) RoadRunner  (42.) Robotank

(43.) Seaquest  (44.) Skiing  (45.) Solaris  (46.) SpaceInvaders  (47.) StarGunner  (48.) Surround

(49.) Tennis  (50.) TimePilot  (51.) Tutankham  (52.) UpNDown  (53.) Venture  (54.) VideoPinball

(55.) WizardOfWor  (56.) YarsRevenge  (57.) Zaxxon

