# OpenReview forum: "CASA: Bridging the Gap between Policy Improvement and Policy Evaluation with Conflict Averse Policy Iteration"
_ICLR.cc/2023/Conference — Submitted to ICLR 2023_

### Official Review · Reviewer_hwhf · 2022-10-22

**Confidence:** 4
**Correctness:** 2
**Technical Novelty And Significance:** 3
**Empirical Novelty And Significance:** 3
**Recommendation:** 5

**Clarity, Quality, Novelty And Reproducibility:**

The writing of the paper is quite suboptimal. Many sentences are not well formulated and make the paper not easily understandable. For example:
- "Since $Q^\pi − Q_\theta$ and $Q^π − V_θ$ are two scalars, when $\nabla_\theta Q_\theta \varpropto \nabla_\theta \log \pi_\theta$, the two sides of $\beta$ meet": instead of saying that the sides meet, considering rephrasing saying that the gradients of policy improvement and policy evaluation overlap;
- "As I and E are perpendicular to each side of β, they become parallel arrows on opposite directions.": how can I and E become arrows? This should be rephrased;
- "By equation 6, the directions of the gradients g are in common. We call it a path. The equation 6 shows that $\nabla_\theta LQ$ and $\nabla_\theta J$ walk along the same path g.": these sentences are redundant and should be merged in one.

The novelty of the proposed method is satisfactory.

The list of used hyperparameters is thoroughly reported for reproducibility, but the source code is missing.

**Strength And Weaknesses:**

Strenghts
-----------
- The paper studies the interesting problem of generalized policy iteration from the original point of view of bridging the gap between policy evaluation and policy improvement;
- Empirical results on Atari are impressive.

Weaknesses
--------------
- The quality of the writing is not satisfactory and makes the paper hardly readable and understandable;
- The proposed method CASA seems to rely on the fact that ALL parameters should be shared between value functions and policy. However, in the paper it is stated that only "most parameters" are shared. Doesn't this critically go against the assumptions of CASA? Please clarify;
- The figures are too small and hardly readable.

**Summary Of The Paper:**

This paper proposes a novel approach to generalized policy iteration (GPI) exploiting the inherent connection between policy evaluation and policy improvement. It firstly shows that the policy evaluation and policy improvement steps are connected in the case of parameter sharing between the value function and the policy, it describes the benefit of this connection and proposes the method CASA to exploit this connection. CASA is tested on the Atari 200M benchmark where it achieves remarkable state-of-the-art results.

**Summary Of The Review:**

This paper studies the important problem of generalized policy iteration in reinforcement learning, and the proposed method is intriguing and able to achieve remarkable results in the challenging Atari 200M benchmark. However, I'm not convinced about the theoretical correctness of the method. It seems to me that conflict between policy evaluation and policy iteration is avoided only in the case where the parameters of value functions and policy are exactly the same. This is never the case in practice. As also stated in the paper, the parameters are mostly shared, except for individual heads. It is not clear to me whether the fact that MOST parameters are shared, is considered sufficiently good to meet the assumption of CASA; in that case, it would be nice to have a better discussion about it. Unfortunately, the low quality of the writing makes everything slightly harder to grasp.

It would be really nice if the authors could solve these doubts in the rebuttal, and I'd gladly increase my score in case the answer will be convincing enough.

Post-rebuttal feedback
------------------------------
I have read the revised paper and appreciate some improvements on the previous version. I am still doubtful about the approach, sharing similar concerns to those expressed by Reviewer aLPu, in particular
> In that sense, this is really a value-based method as there is no "independent" learning of a parameterized policy.

Thus, I raise my score for the improved version but I still consider this work under the threshold of acceptance.

---

> ### Author Response · Authors · 2022-11-14
> **Responses**
>
> Thanks for your suggestions. We update a new version and all responses refer to the new one. Changes are highlighted in red.
>
> ------------------------------------
>
> ### Strength And Weaknesses
>
> - “The quality of the writing is not satisfactory and makes the paper hardly readable and understandable”
>
>     Response: We have revised many paragraphs. We hope it helps.
>
> - “The proposed method CASA seems to rely on the fact that ALL parameters should be shared between value functions and policy. However, in the paper it is stated that only "most parameters" are shared. Doesn't this critically go against the assumptions of CASA? Please clarify;”
>
>     Response: i) Yes, all parameters are shared except for temperature $\tau$. In this work, there is one backbone and two individual heads after the backbone. The advantage function and the policy share one head, and state value function is the other head. Hence the policy reuses all parameters of value functions except that temperature $\tau$ is only for the policy. We rewrite the 1st paragraph in Sec.3.1. ii) The old version writes ‘most’ because temperature $\tau$ is an exception. It can be either trainable or static. In this work, we keep it static.
>
> - “The figures are too small and hardly readable.”
>
>     Response: We have updated Fig.3, Fig.4 with the same y-axis scale. Fig.3 is enlarged. We hope it helps.
>
> ----------------------------------
>
> ### Clarity, Quality, Novelty And Reproducibility
>
> - “"Since Qπ−Qθ and Qπ−Vθ are two scalars ……”&”"As I and E are perpendicular to each side of β……”
>
>     Response: Thanks. We have fixed the paragraph after Eq.4.
>
> - “"By equation 6, the directions of the gradients g are in common……”
>
>     Response: We have fixed the paragraph after Eq.11.
>
> --------------------------------------
>
> ### Summary Of The Review
>
> - “It seems to me that conflict between policy evaluation and policy iteration is avoided only in the case where the parameters of value functions and policy are exactly the same. This is never the case in practice.”
>
>     Response: Thanks for pointing that out. i) We also notice this limitation. In our new version, we formulate that what we want is to make $\cos(\beta)=1$ (defined in Eq.4), and what CASA does is to make $\chi=1$ (defined in Eq.5) in exact. We observe that $\cos(\beta)$ and $\chi$ are positively correlated in ablation experiments. Since $\cos(\beta)$ is hard to be optimized by $\theta$ because of stochasticity and expectation, we instead consider $\chi$, which is fully controllable by $\theta$ for each state. We hope this observation can be extended to non-shared architecture, where $\cos(\beta)$ can be enlarged by increasing $\chi$. ii) We add a simple discussion on this limitation in Sec.6.

---

> > ### Author Response · Authors · 2022-11-30
> > **An Additional Remark**
> >
> > Thanks for your feedback and we appreciate it. We also notice the ‘sharing’ part and make a simple discussion in the 2nd paragraph of Sec.6. We want to expand a little.
> >
> > In general, assume we have $Q_\theta$ and $\pi_\phi$, where $\phi$ can be independent from $\theta$. The definition of CASA is indeed $\log \pi_{casa} = (\alpha*\log \pi_\phi + (1-\alpha)*Q_\theta/\tau)|{\alpha=0}$. This form is similar to MPO$^{[1]}$, where $\log \pi_{mpo} = \log \pi_\phi + Q_\theta / \eta$. We take MPO as an example and compare the two to illustrate CASA is more like a policy-based method while MPO is more like a value-based method. Firstly, the learning signal of $\pi$ is different. MPO distills $\pi_\phi$ to $\pi_{mpo}$, where the signal is $Q_\theta$ i.e. function approximation. CASA learns $\pi$ by policy gradient, where the signal comes from the environment directly. Secondly, MPO also reports ALE in Table.1 in Appendix. The performance is similar to DQN and Rainbow, while CASA is similar to PPO and IMPALA.
> >
> > In conclusion, from the aspect of representation, under the definition of $(\alpha*\log \pi_\phi + (1-\alpha)*Q_\theta / \tau)$, a fully independent $\pi$ corresponds to $\alpha=1$. CASA corresponds to $\alpha=0$. MPO corresponds to $\alpha=0.5, \tau=\eta$. We are still not sure which one is better, as each has its own benefit. The other aspect is the training strategy, which makes CASA policy-based-like and MPO value-based-like.
> >
> > --------------------
> > ## Supplement
> >
> > We further do experiments on a very simple rectangular tabular world. One is the comparison between a fully independent $\pi$, CASA and MPO. Two is the performance of $(\alpha*\log \pi_\phi + (1-\alpha)*A_\theta / \tau)$ with different $\alpha$.
> > This tabular env shows CASA performs the best, without function approximation, hyperparameter tuning and entropy penalty.
> >
> > We release the code at https://github.com/iclr2023casa/iclr2023casa_tabular (under double blind review).
> >
> > --------------------
> >
> > [1] Abdolmaleki, Abbas, et al. "Maximum a posteriori policy optimisation." arXiv preprint arXiv:1806.06920 (2018).

---

### Official Review · Reviewer_aLPu · 2022-10-23

**Confidence:** 3
**Correctness:** 2
**Technical Novelty And Significance:** 3
**Empirical Novelty And Significance:** 2
**Recommendation:** 3

**Clarity, Quality, Novelty And Reproducibility:**

The paper is hard to follow. There are technical details which are not explained well, or are misleading, throughout. Some can be easily fixed, but some are more fundamental.

**Major issues**
- In general, the paper claim to "CASA, Critic AS  an Actor, which theoretically guarantees that the evaluation gradient of the state-action values and the policy  gradient to be parallel". But the way this is achieved is a triviality, because the architecture forces the log-policy to be linear in the state-action values. In that sense, this is really a value-based method as there is no "independent" learning of a parameterized policy. Everything goes through the value (more precisely, advantage) function. And the main idea is to use a Softmax instead of a Max (greedy) for the action-selection rule.

- The definition of $\beta$  and its explanation (Section 3.1) are misleading (or at least unclear). The definitions of $\mathbf{I}$ and $\mathbf{E}$ included (and rightly so) an expectation over state-action pairs as induced by $\pi$. This expectation for some reason is missing from the definition of $\beta$. Note that with this expectation, it is **not** the case that "$Q^\pi-Q_\theta$  and $Q^\pi-V_\theta$  are two scalars" that can be pulled out of the expectation because in general they will be correlated with the gradients. In fact the entire PG method relies on this correlation, otherwise the expectation value of the score function by itself $\mathbb{E}\left[\nabla\log\pi\right]$ is simply $0$. Since this is not explained it's also unclear how $\beta$ is calculated empirically later, from full trajectories.
- The explanation that follows is also confusing, or at least I couldn't understand what is being claimed (The math says that $\mathbf{I}$ and $\mathbf{E}$ are parallel, but the text says they are perpendicular?)
-  Related to that, the question on "under what conditions $\nabla_\theta Q_\theta \propto \nabla_\theta\log\pi$  is misleading because $\nabla_\theta Q_\theta$ does not reflect policy evaluation at all, if anything it is more related to policy improvement: since $\theta$ (implicitly) parameterizes the policy, $\nabla_\theta Q_\theta$  (implicitly) means a change in the policy that will make the action values higher, i.e., improve the policy.

- The fact that all experiments are done with a recurrent network (a detail that is somewhat "hidden" in Section 4.1) is a very serious shortcoming for the evaluation. The entire theoretical attempt is built on top of memoryless/reactive value-function and policy estimations (as is standard for MDPs), but then the experiments is done with an RNN. This might be a source of correlations in the gradients, estimations, etc. throughout the trajectories which is completely out of scope of the more "theoretical" explanation.

**Minor issues**
*(or, really, not minor but more easily fixable)*

*Preliminary section:*
- The presented objective wrongly mixes the "trajectory" and "state-action occupancy measure" formulations (e.g. Sutton et al. 1999, Ghosh et al. 2020) for the RL objective. If $s\sim d^\pi$  then this already encapsulates the $\sum_t \gamma^t \mathbb{P}^\left(t\right)\left[s\right]$ averaging. Since the discounted occupancy measure is not really used later in the paper it seems preferable to simply write the objective in the "trajectory formulation" i.e., $J=\sum_t\left[\gamma^tr\left(s_t,a_T\right)\right]$ and explain how the sequence $s_t,a_t$ is being sampled by the policy-environment interaction. Moreover:
	- it is currently not defined that $a_t$ is sampled via $\pi\left(s_t,\cdot\right)$ .
	- The measure $d^\pi$ as defined is **NOT** the "stationary distribution induced by $\pi$" but rather the normalized discounted occupancy measure / visitation distribution. It becomes the stationary distribution only in the limit $\gamma=1$ (yielding the time-average criterion instead of the discounted criterion of performance).

- In the next paragraph, the definition of $V\left(s\right)$ and $Q\left(s,a\right)$ should be conditioned on $s_0=s$ and $s_0=s,a_0=a$ respectively, rather than on $s$ and $s,a$ which can be confusing to interpret.

- The same issue of mixing the "trajectory" and "state-action occupancy measure" repeats in the paragraph on Policy-Gradient methods (namely $\Psi$ has an explicit time-index inside it which is not really defined. Contrast this to how it is defined in Schulam et al. 2015, Eq.1).
- It is not clear why the concept of $\gamma$-justness, which is never used again in the paper, is introduced here, instead of a more simple explanation. This also confuses between the exact PG and an estimate of it based on sampled trajectories (the concept of  ""$\gamma$-justness" being relevant to the latter rather than the former case, while here the paper describes the exact gradient).

- The Equations defining the objective/optimization problems, gradients, and (in the next Section) $\beta$ should all be numbered.

*Methodology section*:
- Why is the PG theorem cited *again*, only a few sentences after it was already presented, but using a different form than the one presented in the Preliminary section? This will only confuse readers.

- What does $\mathbb{E}_p$  represents (in another un-numbered equation, discussing the relation to dueling DQN)? what is $p$?
- What is $\mu$ in Table 1?

- The text in Figure 3 (axes labels, labels) is **tiny** and becomes readable only in 350% zoom.

- Figures 4 and 3: Several panels report $\cos$ values side by side but have (very) different y-axis scale. This is very misleading. All Figures depicting $\cos$ values should have a normalized scale between 0 to 1 (or -1 to 1 if relevant).

*Related work*:
- In general, the concept of $\beta$ might be related to the concept of a "compatible value function" (Sutton et al. 1999, see also Kakade 2001). It would be interesting if the proposed concept here can (maybe after some corrections/modifications) serves as a generalization of the original compatibility notion (which was more of a binary concept, yes or no). I would encourage the authors to relate and explain how and if their ideas relate to that, or at the very least mention it in the related work.

**Strength And Weaknesses:**

The basic ideas underlying the paper are interesting and could potentially be a valid contribution. However there are several key issues that make the claims either difficult to understand (in the good case) or wrong/inaccurate (in the worse case). The evaluation is also not satisfying, relying on complex architectures and environments (see more below) to try and support what are fundamentally abstract theoretical claims.

I would recommend the authors to reconsider the evaluation experiments that are used to support the theoretical claims. Ultimately, even a much simpler tabular settings where everything is much more controlled can be a much better starting point for this kind of paper, even if not all claims or properties can be rigorously proven theoretically.

**Summary Of The Paper:**

This works studies the relation between policy evaluation and policy improvement in the context where the value-function(s) and the policy share the same parameterization, and suggests a way to consolidates the two type of updates.

**Summary Of The Review:**

The paper deals with an interesting question and some parts or aspects of it could be a valid contribution. However in its current form it has too many open gaps and issues that should be fixed before it is ready for publication.

---

> ### Author Response · Authors · 2022-11-14
> **Responses**
>
> Thanks for your suggestions. We update a new version and all responses refer to the new one. Changes are highlighted in red.
>
> ------------------------------------
>
> ### Major issues
>
> - “In general, the paper claim to "CASA, Critic AS an Actor, which theoretically guarantees that the evaluation gradient of the state-action values and the policy gradient to be parallel". But the way this is achieved is a triviality, because the architecture forces the log-policy to be linear in the state-action values. In that sense, this is really a value-based method as there is no "independent" learning of a parameterized policy. Everything goes through the value (more precisely, advantage) function. And the main idea is to use a Softmax instead of a Max (greedy) for the action-selection rule”
>
>     Response: i) We acknowledge that the technical way is simple. But we refer to the two sg operators, which are explainable and discussed after Eq.6. They are crucial to guarantee $\chi=1$ (defined in Eq.5), which is a weaker compatible condition (discussed after Eq.5 and in Appendix.A). ii) We add a simple discussion about this limitation in Sec.6.
>
> - “The definition of β and its explanation (Section 3.1) are misleading (or at least unclear). The definitions of I and E included (and rightly so) an expectation over state-action pairs as induced by π. This expectation for some reason is missing from the definition of β. Note that with this expectation, it is not the case that "Qπ−Qθ and Qπ−Vθ are two scalars" that can be pulled out of the expectation because in general they will be correlated with the gradients. In fact the entire PG method relies on this correlation, otherwise the expectation value of the score function by itself E[∇log⁡π] is simply 0. Since this is not explained it's also unclear how β is calculated empirically later, from full trajectories.”
>
>     Response: i)  We add accurate definitions of $\beta$ (defined in Eq.4) and $\chi$ (defined in Eq.5). In summary, $\beta$ takes expectation first, and $\chi$ calculates angle for each state first. ii) We add 2nd paragraph in Sec.4.1 and 3rd paragraph in Appendix.B to carefully describe our experimental settings. The expectation operation is batch-wise average, which each batch contains (batch size * seq length) states.
>
> - “The explanation that follows is also confusing, or at least I couldn't understand what is being claimed (The math says that I and E are parallel, but the text says they are perpendicular?)”
>
>     Response: The $\beta$ angle in old Fig.1 and old Fig.2 is inaccurate. The description is about old Fig.1 so it’s misleading. We replace figures and re-write descriptions(after Eq.4).
>
> - “Related to that, the question on "under what conditions ∇θQθ∝∇θlog⁡π is misleading because ∇θQθ does not reflect policy evaluation at all, if anything it is more related to policy improvement: since θ (implicitly) parameterizes the policy, ∇θQθ (implicitly) means a change in the policy that will make the action values higher, i.e., improve the policy.”
>
>     Response: Thanks for pointing that out. We define $\beta$ (Eq.4) and $\chi$ (Eq.5) and they are different. The idea is that it’s hard to let $\beta=0$ by varying parameters $\theta$ as it’s intractable with expectation and stochasticity inside. So we remove step sizes and take expectation outside, which is $\chi$. We observe that $\cos(\beta)$ is closer to 1 as $\chi$ is closer to 1. So we want to let $\chi=1$, as $\chi$ is fully controlled by $\theta$ for each state.
>
> - “The fact that all experiments are done with a recurrent network (a detail that is somewhat "hidden" in Section 4.1) is a very serious shortcoming for the evaluation. The entire theoretical attempt is built on top of memoryless/reactive value-function and policy estimations (as is standard for MDPs), but then the experiments is done with an RNN. This might be a source of correlations in the gradients, estimations, etc. throughout the trajectories which is completely out of scope of the more "theoretical" explanation.”
>
>     Response: i) We re-run all experiments in motivation (Sec.3.1) and ablation (Sec.4.2, Sec.4.3) parts with a new backbone IMPALA,shallow without LSTM. Burn-in is also removed. Hyperparameters in Appendix.F are changed. ii) Only experiment on 57 atari games (Sec.4.4) uses IMPALA,deep, as it’s compared with IMPALA so the same backbone is used.

---

> > ### Author Response · Authors · 2022-11-14
> > **Responses 2**
> >
> > ------------------------------------------
> >
> > ### Minor issues Methodology section
> >
> > - “Why is the PG theorem cited again……”
> >
> >     Response: It’s a typo. We remove it.
> >
> > - “What does Ep represents……”
> >
> >     Response: It’s the transition probability distribution of the environment. It’s unnecessary, so we remove it and re-write the last paragraph of Sec.3.2.
> >
> > - “What is μ in Table 1?”
> >
> >     Response: It’s the behavior policy. We add description at the beginning of Sec.3.4 and Appendix.D.
> >
> > - “The text in Figure 3 (axes labels, labels) is tiny and becomes readable only in 350% zoom.”
> >
> >     Response: We have updated Fig.3 and hope it helps.
> >
> > - “Figures 4 and 3: Several panels report cos values side by side but have (very) different y-axis scale. This is very misleading. All Figures depicting cos values should have a normalized scale between 0 to 1 (or -1 to 1 if relevant).”
> >
> >     Response: We have updated Fig.3, Fig.4 and figures in Appendix.B with the same y-axis scale.
> >
> > ----------------------------------------------
> >
> > ### Minor issues Related work
> >
> > - “In general, the concept of β might be related to the concept of a "compatible value function"......”
> >
> >     Response: Thanks for pointing that out. i) Yes, it’s very interesting to see that CASA satisfies a weaker compatible condition ($\chi=1$ in exact). According to our observation, $\cos(\beta)$ (defined in Eq.4) and $\chi$ (defined in Eq.5) show positive correlation. And $\chi=1$ is a necessary condition of the compatible condition (paragraph after Eq.5 and Thm.3 in Appendix.A). ii) We also add this at the end of related works (Sec.5).

---

> > > ### Author Response · Authors · 2022-12-03
> > > **Supplement**
> > >
> > > We further do experiments on a very simple rectangular tabular world. One is the comparison between a fully independent $\pi$, CASA and MPO. Two is the performance of $(\alpha*\log \pi_\phi + (1-\alpha)*A_\theta / \tau)$ with different $\alpha$.
> > > This tabular env shows CASA performs the best, without function approximation, hyperparameter tuning and entropy penalty.
> > >
> > > We release the code at https://github.com/iclr2023casa/iclr2023casa_tabular (under double blind review).
> > >
> > > The discussion about 'sharing' (representation) part, and the comparison between the training strategy of CASA and MPO can be found in $\textbf{An Additional Remark}$ for Reviewer hwhf.

---

### Official Review · Reviewer_QNNj · 2022-10-23

**Confidence:** 3
**Correctness:** 3
**Technical Novelty And Significance:** 1
**Empirical Novelty And Significance:** 1
**Recommendation:** 3

**Clarity, Quality, Novelty And Reproducibility:**

The paper has a rather scattered presentation style. I found it hard to follow the storyline, disambiguate the studied problem, and a main hypothesis drawn as a solution to it. This also makes it hard to view the experiment results as empirical evidence to a scientific claim.

**Strength And Weaknesses:**

*Strengths:*

The formulation of the beta variable is interesting.

*Weaknesses:*

 * I cannot make sense of the studied problem at all. What does it mean to have a gradient conflict between two operational steps (policy evaluation and policy improvement) which have complementary contributions to the solution?
 * The paper states its aim as to “eliminate the inconsistency between the policy evaluation and policy improvement steps”. However, I do not understand how the proposed CASA method solves this problem. It only sets up a tunable continuum between these two steps. But one would also get the same using the \lambda-policy iteration formula, which is introductory textbook stuff.
 * I do not find the reported ablation study informative. The benefit of CASA can be more directly quantified if the paper compares PPO vs PPO+CASA and R2D2 vs R2D2+CASA directly in the Atari game use case.
 * Overall the experiments section is rather slim. The paper does not compare against some strong candidates such as SAC and MBPO.


**Summary Of The Paper:**

The paper studies the problem of solving the alleged inconsistencies between the policy evaluation and policy improvement steps of the “policy iteration” algorithm. The paper proposes a method called CASA as a solution which suggests some modifications on the gradient calculations of standard algorithms such as PPO.

**Summary Of The Review:**

The studied problem of the paper is not stated in a technically rigorous and unambiguous and intuitively clearly motivated way. The solution attempt also does not appear to follow a consistent deductive process. The reported experiment results have weak relevance to the main hypothesis.

---

> ### Author Response · Authors · 2022-11-14
> **Responses**
>
> Thanks for your suggestions. We update a new version and all responses refer to the new one. Changes are highlighted in red.
>
> -------------------------------------------
>
> ### Strength And Weaknesses
>
> - “I cannot make sense of the studied problem at all. What does it mean to have a gradient conflict between two operational steps (policy evaluation and policy improvement) which have complementary contributions to the solution?”
>
>     Response: For each policy evaluation/improvement step, the parameters should move toward the calculated gradient of the policy evaluation/improvement step. But when they are trained concurrently and both gradients are applied, the gradient of the policy evaluation/improvement step will be influenced by the gradient of the policy improvement/evaluation step, which is not the original direction.
>     Then the policy evaluation/improvement step may not be updated as it should be. So we are looking for a solution where the gradients are not influenced much by each other, which is to be parallel (or consistent in the paper).
>
> - “The paper states its aim as to “eliminate the inconsistency between the policy evaluation and policy improvement steps”. However, I do not understand how the proposed CASA method solves this problem. It only sets up a tunable continuum between these two steps. But one would also get the same using the \lambda-policy iteration formula, which is introductory textbook stuff.”
>
>     Response: As it is hard to let $\beta=0$ (defined in Eq.4) by varying parameter $\theta$, we instead consider how we can make $\chi=1$ (defined in Eq.5), which is also a weaker compatible condition (discussed after Eq.5 and in Appendix.A). CASA lets $\chi=1$ in exact. It’s not just adding a tunable temperature, but two stop gradient operators are crucial and explained after Eq.6.
>     Experiments show that $\cos(\beta)$ is closer to 1 as $\chi$ is closer to 1.We observe that $\chi$ and $\cos(\beta)$ are positively correlated. This is how we make $\cos(\beta)$ closer to 1.
>
> - “I do not find the reported ablation study informative. The benefit of CASA can be more directly quantified if the paper compares PPO vs PPO+CASA and R2D2 vs R2D2+CASA directly in the Atari game use case.”
>
>     Response: Returns curves are already shown in Fig.3 and Fig.4. All curves in figures are distinguishable. The motivation and ablation study are designed more to understand the angles of gradients between different losses ($\beta$ defined in Eq.4) and also the compatible condition ($\chi$ defined in Eq.5). We give more concentration on the behavior of angles in-between rather than the simple scores. All scores and return curves of 57 atari games are in Table.4 and Appendix.G.
>
> - “Overall the experiments section is rather slim. The paper does not compare against some strong candidates such as SAC and MBPO.”
>
>     Response: i) Rainbow and IMPALA are strong model-free baselines on the 200M Atari 57 games benchmark, and 200M Atari 57 games benchmark is a strong benchmark. ii) We have not found an official reported result of SAC on the 200M Atari 57 games benchmark, so we cannot make a fair comparison. iii) MBPO adopts a model-based setting, which is not a proper baseline, as the entire paper talks about model-free algorithms.

---

> > ### Comment · Reviewer_QNNj · 2022-11-28
> > **Keep my score**
> >
> > The author response does not address my concerns. I do not think not finding any prior work on a SOTA algorithm would be grounds for avoiding comparison to it. One can simply make the benchmarking study oneself, otherwise the claims made about the advancement of the SOTA are in vain.

---

> > > ### Author Response · Authors · 2022-11-30
> > > **A Feedback**
> > >
> > > Our work is doing model-free RL, so we don’t think we have to compare with MBPO.
> > >
> > > As for the SAC, we have already explained that for the 200M Atari benchmark, Rainbow, IMPALA and LAZER are strong and comparable baselines to our work. We do implement SAC$^{[1]}$, but for this 200M Atari benchmark, the result of SAC is far less than ours, Rainbow’s and IMPALA’s, even though we have tuned the target entropy many times. We know any implementation detail may cause different results. We do not want to under/over-estimate any algorithm, unless the algorithm officially reports the result on the 200M Atari benchmark.
> > >
> > > Another side proof is this paper$^{[2]}$, which implements SAC on 20 Atari games. The result in Appendix is only comparable to Rainbow’s and less than ours.
> > >
> > > We hope to see more insightful and interesting ideas, rather than simply complaining about the comparison (even if we have already compared with strong baselines of model-free methods on the 200M Atari benchmark).
> > >
> > > In conclusion, we do NOT acknowledge the quality of the review and the reason to keep the score.
> > >
> > > -----------------
> > >
> > > [1] Haarnoja, Tuomas, et al. "Soft actor-critic algorithms and applications." arXiv preprint arXiv:1812.05905 (2018).
> > >
> > > [2] Christodoulou, Petros. "Soft actor-critic for discrete action settings." arXiv preprint arXiv:1910.07207 (2019).

---

### Official Review · Reviewer_o19T · 2022-11-03

**Confidence:** 4
**Correctness:** 4
**Technical Novelty And Significance:** 4
**Empirical Novelty And Significance:** 3
**Recommendation:** 6

**Clarity, Quality, Novelty And Reproducibility:**

The writing of the paper is mostly clear, but could be improved in places.

In the Preliminary section, the objective $J$ is written incorrectly. If you’re taking the expectation over the discounted stationary distribution, the random variable inside the expectation should just be $r(s,a)$, not the discounted sum of rewards.

The CASA approach is novel, to my knowledge.

The appendix includes the relevant Python package versions and an exhaustive list of hyperparameter settings, so I believe it would be reproducible.

**Strength And Weaknesses:**

Strengths:
* The paper addresses an issue that many ignore in deep RL – the relationship between the policy evaluation and policy improvement steps in policy iteration.
* The clever usage of sg in the objective leads to the desired result without any approximation or extra computationally intensive steps.
* CASA substantially improves the performance of the base algorithm (IMPALA) on the Atari benchmark.

Weaknesses:
* The method as presented only applies to discrete-action problems, owing to the softmax parameterization of the policy and the summation over actions when computing the advantage.
* The paper shows plots for many different gradient angles, but it’s not clear which we should care the most about. It would be helpful to have a better understanding of how the angles between the gradients correlate with performance.

**Summary Of The Paper:**

The paper presents a method for reducing gradient interference between different components (Q, V, and policy) in deep RL with discrete actions. By carefully placing stop-gradient (sg) operators in the objective, the authors show (both theoretically and empirically) that the gradients between the Q function and the policy can be made parallel, so that there is no interference. The method, CASA, can be applied to value-based or policy-based algorithms using the same computations, and attains strong performance on Atari.

**Summary Of The Review:**

I think the paper is a useful contribution, as it tackles a relatively underexplored problem with a clever and effective solution.

---

> ### Author Response · Authors · 2022-11-14
> **Responses**
>
> Thanks for your suggestions. We update a new version and all responses refer to the new one. Changes are highlighted in red.
>
> ------------------------------------
>
> ### Strength And Weaknesses
>
> - “The method as presented only applies to discrete-action problems, owing to the softmax parameterization of the policy and the summation over actions when computing the advantage.”
>
>     Response: We add a discussion about the limitation in Sec.6, and add Appendix. C to discuss applying CASA on continuous action space.
>
> - “The paper shows plots for many different gradient angles, but it’s not clear which we should care the most about. It would be helpful to have a better understanding of how the angles between the gradients correlate with performance.”
>
>     Response: We highlight eq. 4 and eq. 5 in Sec.3.1. They are the most important quantity of this work. Eq.4 represents gradient consistency between policy improvement and policy evaluation, where the gradient is more consistent as $\cos(\beta)$ is closer to 1. Eq.5 is a more controllable variant of Eq.4. CASA satisfies $\chi=1$ (defined in Eq.5), and shows higher $\cos(\beta)$ and better performance. Hence, we expect $\chi$ and $\cos(\beta)$ to be positively correlated.
>
> -------------------------------------------
>
> ### Clarity, Quality, Novelty And Reproducibility
> - “In the Preliminary section, the objective J is written incorrectly.”
>
>     Response: Thanks. We fix Eq.1 and the following paragraph.

---

### Decision · Program_Chairs · 2023-01-20

**Decision:**

Reject

**Justification For Why Not Higher Score:**

The paper needs to be improved from several points of view.
The presentation is poor and the significance of the contribution is quite limited given the assumptions considered.

**Justification For Why Not Lower Score:**

N/A

**Metareview: Summary, Strengths And Weaknesses:**

The paper studies the Generalized Policy Iteration approach when the same parameterization is shared between the policy and the value function. The authors propose a method to reduce the interference between the updates of the different components and test it on Atari games.
After reading each others' reviews and the authors' feedback, the reviewers acknowledge that the paper has merits (relevant topic, interesting approach, good empirical results), but they raise some concerns (the description of key concepts is not clear, the presentation is poor, the significance of the contribution is quite limited) that need to be solved before the paper can be accepted for publication.
We encourage the authors to consider the reviewers' suggestions while preparing a new version of their paper.


**Summary Of Ac-Reviewer Meeting:**

N/A